# Layer-by-layer self-assembly of pillared two-dimensional multilayers

Weiqian Tian [1,2,4], Armin VahidMohammadi [3,4], Zhen Wang[1,2], Liangqi Ouyang[1], Majid Beidaghi[3] & Mahiar M. Hamedi [1,2]

We report Layer-by-Layer (LbL) self-assembly of pillared two-dimensional (2D) multilayers, from water, onto a wide range of substrates. This LbL method uses a small molecule, tris(2-aminoethyl) amine (TAEA), and a colloidal dispersion of $Ti_3C_2T_x$ MXene to LbL self-assemble $(MXene/TAEA)_n$ multilayers, where n denotes the number of bilayers. Assembly with TAEA results in highly ordered $(MXene/TAEA)_n$ multilayers where the TAEA expands the interlayer spacing of MXene flakes by only ~ 1 Å and reinforces the interconnection between them. The TAEA-pillared MXene multilayers show the highest electronic conductivity of $7.3 \times 10^4$ S m$^{-1}$ compared with all reported MXene multilayers fabricated by LbL technique. The $(MXene/TAEA)_n$ multilayers could be used as electrodes for flexible all-solid-state supercapacitors delivering a high volumetric capacitance of 583 F cm$^{-3}$ and high energy and power densities of 3.0 Wh L$^{-1}$ and 4400 W L$^{-1}$, respectively. This strategy enables large-scale fabrication of highly conductive pillared MXene multilayers, and potentially fabrication of other 2D heterostructures.

[1] Department of Fibre and Polymer Technology, KTH Royal Institute of Technology, Teknikringen 56, 10044 Stockholm, Sweden. [2] Wallenberg Wood Science Centre, Department of Fibre and Polymer Technology, KTH Royal Institute of Technology, Teknikringen 56, 10044 Stockholm, Sweden. [3] Department of Mechanical and Materials Engineering, Auburn University, Auburn, AL 36849, USA. [4] These authors contributed equally: Weiqian Tian, Armin VahidMohammadi. Correspondence and requests for materials should be addressed to M.B. (email: mbeidaghi@auburn.edu) or to M.M.H. (email: mahiar@kth.se)

Two-dimensional (2D) materials are rapidly emerging as a class of materials and their most interesting applications arise when single layers of one or several 2D materials are stacked to form pillared 2D multilayers[1–3] or Van der Waals heterostructures[4]. The large-scale fabrication of such structures, however, remains an unsolved challenge. Recently a number of experimental[5] and theoretical[6] works have shown that thousands, and potentially many more, 2D materials can be exfoliated to form stable dispersions of 2D sheets. Exfoliated 2D sheets, especially in water, present a great opportunity for self-assembly of advanced 2D pillard or hetero-layered structures. Here we use 2D $Ti_3C_2T_x$, a member of 2D transition metal carbides and nitrides called MXenes[7]—a family of materials that have shown great promise for numerous applications[8–15]—to form 2D multilayers using an aqueous layer-by-layer (LbL) self-assembly technique. This method allows single flake assembly precision in each layer and sub-nanometer precision of the interlayer spacing.

MXenes are usually produced by a selective etching process in which "A" layer atoms of MAX phases, a large group of layered ternary carbides and nitrides, are removed in fluoride containing acidic solutions[16,17]. The aqueous etching and exfoliation process results in surface functionalized MXenes with a general formula of $M_{n+1}X_nT_x$, where M is a transition metal, X is carbon/nitrogen, n can be 1–3, and $T_x$ represents the different O, OH, and F surface terminations[8]. The unique combination of metallic conductivity and functionalized surfaces has rendered MXenes as potential candidates for fast energy storage electrodes, because the metal carbide layers provide excellent charge transfer inside the electrodes and functionalized surfaces enhance the pseudocapacitive response of MXene electrodes[18,19].

Freestanding MXene films, fabricated by vacuum filtration of MXene dispersions consist of random stacks of the delaminated flakes and show poor resistance to strain because of the sliding between the stacked MXene flakes during the mechanical deformations[20]. In addition, inevitable self-restacking of MXene flakes in such structures reduces the accessibility of electrolyte's ions to the interior redox-active surfaces of MXenes thereby impeding the ionic transport channels for charge storage[18,21]. These issues have largely prevented full exploration of the charge storage capability of MXenes. Luckily, the high-aspect-ratio structure of MXene flakes and their highly functionalized surfaces enables them to be assembled into multifunctional pillared structures using LbL self-assembly[1]. LbL is typically a cyclical process in which two oppositely charged species are alternately deposited onto a substrate to form multilayer structures with a thickness that scales with the number of layers[22]. In LbL architectures, the different layers are held together by electrostatic interactions, covalent and hydrogen bonding, or ionic charge transfer[1]. LbL self-assembly could allow the different functional components—structural, ion conducting, and electron conducting phases—to be optimally arranged at the nanoscale and lead to an integrated network with a better interfacial strength, higher conductivity, and more accessible active surfaces[23].

Recently, LbL self-assembly technique has been employed to assemble MXene multilayers with polyelectrolytes such as polyethyleneimine (PEI)[24], poly(diallyldimethylammonium chloride) (PDAC)[25], or poly(sodium 4-styrene sulfonate)[26]/PEI-modified carbon nanotube[27]. The electrically insulating polymers in these multilayers, however, form large gaps between the adjacent MXene flakes which disrupt the electron conduction paths, thus preventing the formation of MXene-based LbL architectures with high electrical conductivity. The polymers are far larger than the thickness of individual MXene flakes, and therefore these systems do not enable LbL self-assembly with a single/few flake precision[25]. The addition of inactive polymers further increases the weight and volume of electrochemically inactive components in

the electrodes[28]. It is thus still a challenge to achieve a highly conductive LbL structure using MXene or other 2D materials with single/few flakes precision in each layer, and with a small gap between the 2D layers.

Here, we solve this problem by introducing a positively-charged triamino, small molecule, tris(2-aminoethyl) amine (TAEA), as the interlayer pillaring component for the LbL self-assembly of $Ti_3C_2T_x$ MXene, to fabricate pillared multilayers of $(MXene/TAEA)_n$ in which the MXene flakes are assembled in a face-to-face quasi-intimate contact leading to a high packing density. The anchored pillars of TAEA create a small gap between the interface contact of MXene flakes resulting in a high electronic conductivity, and in a slightly expanded interlayer spacing between the individual MXene flakes which accelerates ion diffusion and provides facile access to the titanium atoms at the surface of MXene layers for fast pseudocapacitive charge storage. Benefitting from those features, the $(MXene/TAEA)_n$ multilayers show excellent electrochemical performance when used as supercapacitor electrodes, and they can also be self-assembled onto various non-metallic substrates including planar films, nonwoven fabrics, paper, foams, and even aerogels. We demonstrate that these multilayer films are conformal, conductive, and resilient to bending and compression.

## Results

**LbL assembly of $(MXene/TAEA)_n$ multilayers.** We fabricated the $(MXene/TAEA)_n$ multilayer films (Fig. 1a) by sequential LbL deposition of positively-charged TAEA, and aqueous dispersions of negatively-charged $Ti_3C_2T_x$ MXene flakes—produced by selective etching of Al atoms from $Ti_3AlC_2$ MAX phase in a solution of HCl and LiF mixture (see Methods and Supplementary Figs. 1, 2). The LbL films are denoted $(MXene/TAEA)_n$ where n corresponds to the number of steps that are repeated in the LbL deposition process.

Atomic force microscope (AFM) images showed that the thickness of individual flakes was around 2.5 nm, and their lateral size distribution was in the range of several hundreds of nanometers (Fig. 1b–d). The higher measured thickness of $Ti_3C_2T_x$ compared to its theoretical thickness (0.98 nm) is in agreement with literature reports and is believed to result from the confinement of water molecules[29].

To optimize the LbL process, we adjusted the pH values of the $Ti_3C_2T_x$ colloidal dispersion and the TAEA solution, and monitored the changes in the zeta potential of MXene and in the charge density of TAEA (Fig. 2a). For the MXene solution, we chose a colloidally stable condition with a zeta potential of −35 mV at pH = 6.5. For the TAEA solution, we achieved a high charge density of +25.2 μeq mL$^{-1}$ at pH = 7.5 which is significantly higher than that of PEI (+18.9 μeq mL$^{-1}$) under the same conditions, due to the protonation of all primary amino groups of TAEA molecule[30,31]. The increased charge density is in favor of the self-assembly processes (Supplementary Fig. 3), especially, LbL self-assembly as it provides stronger electrostatic interactions between TAEA and MXene. In addition, the protonated amino groups of the TAEA have a high affinity with the terminal group-doped metallic surfaces of MXene flakes and may even form chemical bonds[32,33]. We also used other small amino molecules such as diamino molecule of spermidine, and slightly larger triamino molecule of tris(3-aminopropyl)amine (TAPA) as the counter phase for the LbL self-assembly of MXene (Supplementary Fig. 4). The results showed that using a triamino molecule is a requirement for achieving LbL self-assembly of highly ordered multilayers, and importantly the TAEA chosen is the smallest size of triamino molecules which can be chosen for our LbL process.

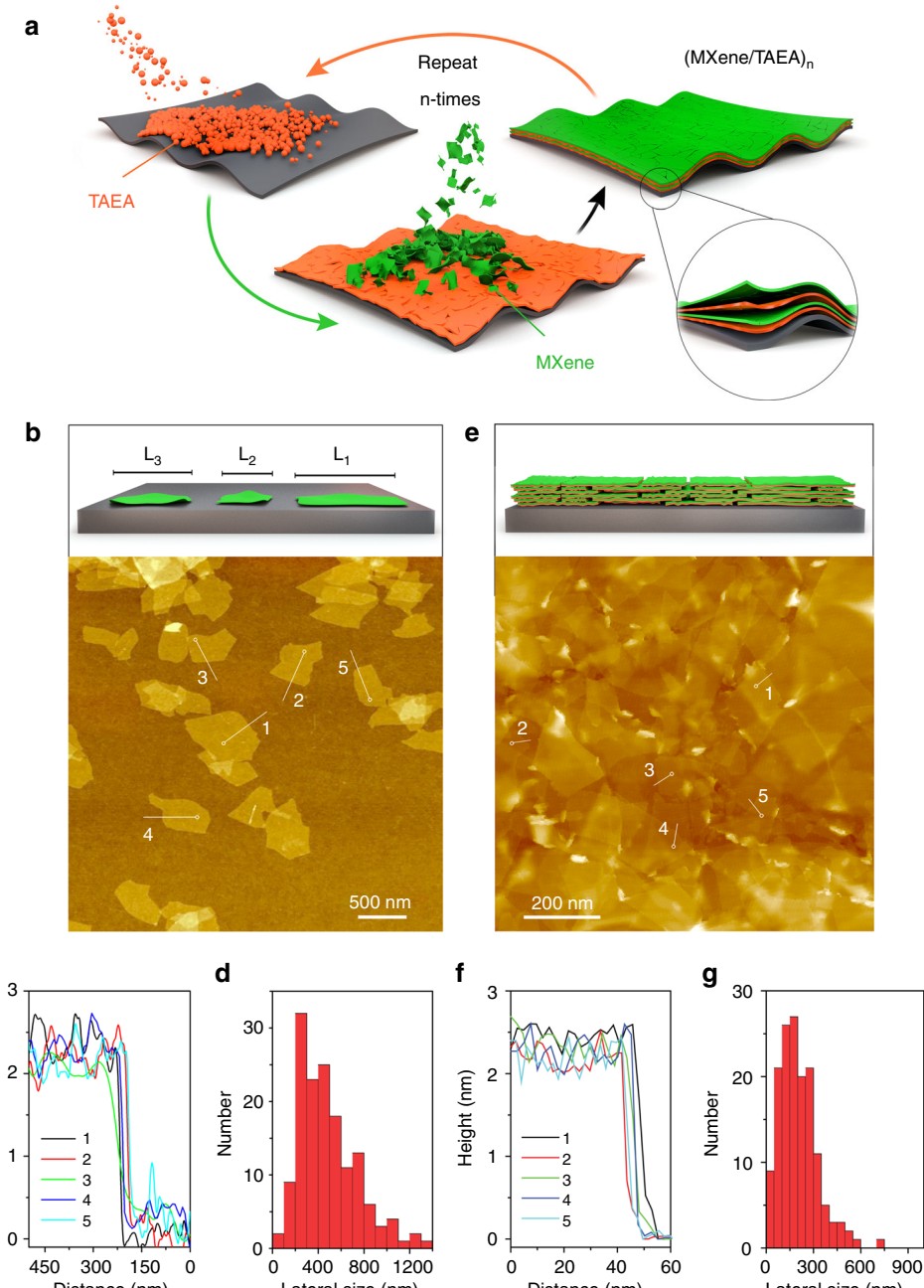

**Fig. 1** Morphological characterization of $Ti_3C_2T_x$ MXene and $(MXene/TAEA)_n$ multilayers. **a** Schematic illustration of the LbL self-assembly of $(MXene/TAEA)_n$ multilayer films onto planar substrates. **b** AFM image of delaminated MXene flakes on a silicon wafer, and **c** corresponding height profiles at the edge of individual MXene flakes in (**b**). **d** The histogram of the lateral size of MXene flakes based on statistics from 150 individual flakes measured from AFM images. **e** AFM image of $(MXene/TAEA)_6$ on a silicon wafer, and **f** corresponding height profiles at the edge of MXene flakes on the top layer of multilayers in (**e**) that were stacked on the next TAEA layer. **g** The histogram of the lateral size of the MXene flakes of $(MXene/TAEA)_6$ based on the statistic of 150 individual flakes from AFM images

To investigate the growth behavior of LbL MXene multilayer films, we used a spin-assisted immersive-LbL self-assembly technique to grow the $(MXene/TAEA)_n$ on planar silicon wafers, polyethylene terephthalate (PET) films, and glass slides (see Methods for details and Fig. 2f). We measured the thickness of $(MXene/TAEA)_n$ multilayers from the cross-sectional SEM images (Fig. 2b and Supplementary Fig. 6), which showed a linear increase in thickness with the bilayer number "n". The mass loading of the multilayers also increased linearly with n (Fig. 2c). The linear behavior is a feature of a successful LbL self-assembly[34], and indicates that the two different materials completely alternate during the LbL process (Supplementary Fig. 11). We further used AFM to analyze the in-plane microstructure of the LbL films (Fig. 1e, and Supplementary Figs. 7, 8). The AFM images showed that all individual MXene flakes in multilayer films stacked face-to-face, in agreement with the top-view SEM images (Supplementary Fig. 5), with a very small arithmetical-mean-deviation roughness $R_a$ of 2.45 nm in an area of 5 μm × 5 μm (Supplementary Fig. 7). AFM images show that the lateral size distribution of the MXene flakes in the multilayer is below 600 nm (Fig. 1g). Figure 1f shows that the edge height of individual MXene flakes on the top layer of

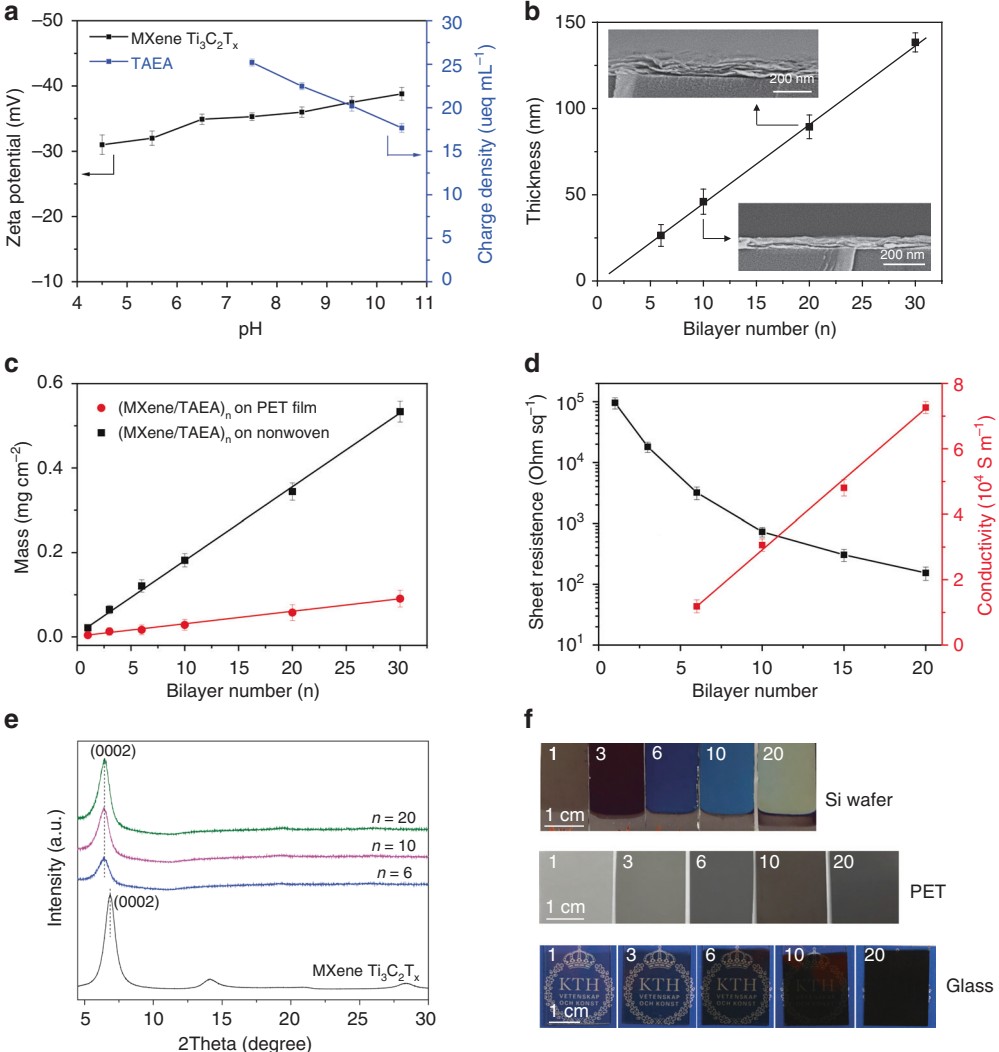

**Fig. 2** Characterization of (MXene/TAEA)$_n$ multilayers. **a** Zeta potential of Ti$_3$C$_2$T$_x$ MXene colloidal solution and charge density of TAEA solution as a function of pH. **b** The average thickness of (MXene/TAEA)$_n$ measured from SEM cross-section images *vs.* bilayer number n. The inset **b** is the corresponding cross-sectional SEM images of (MXene/TAEA)$_n$ on silicon wafers. **c** The mass loading of (MXene/TAEA)$_n$ onto PET sheets and nonwoven fibers vs. n. **d** Sheet resistance and electric conductivity of (MXene/TAEA)$_n$ on silicon wafers vs. n. The data in **a**–**e** show statistical values from 6 measurements in each data point. **e** XRD patterns of (MXene/TAEA)$_n$ vs. n, and pure MXene Ti$_3$C$_2$T$_x$ film. **f** Digital photographs of the (MXene/TAEA)$_n$ on silicon wafers, PET sheets and glass slides vs. n. The color change on Si wafers is owing to different optical characteristics of various thickness of (MXene/TAEA)$_n$

multilayers is ~2.5 nm, which is consistent with the thickness of individual flakes of pristine Ti$_3$C$_2$T$_x$ MXene (Fig. 1c). This suggested that individual MXene flakes did not agglomerate during LbL process. We propose that the uniform and face-to-face deposition of single MXene flakes results from the spin-assisted process which provides the shear force at the interface between MXene flakes and the substrate surface to prevent the deposition of large flakes and formation of thick agglomerates[35].

XRD patterns of the (MXene/TAEA)$_n$ multilayers (Fig. 2e) showed ordered structures similar to pristine MXene films[36,37], and the intensity of (0002) peaks, stemmed from the ordered stacking of MXene flakes' basal planes, increased with the number of bilayers, n. We note that a more ordered and smooth LbL structure was obtained only when the counter phase was the small-molecule TAEA, compared with those of polymer-based LbL structures such as (MXene/PEI)$_n$ (Supplementary Fig. 9) and the reported (MXene/PDAC)$_n$[25] and (MXene-PVA/CNT-PSS)$_n$[26] films whose (0002) peak totally disappeared due to their less ordered structure. We attribute the ordered structure to the small

size of TAEA which forms a sub-nanometer gap in between MXene flakes in the LbL films, leading to a quasi-intimate interfacial contact between the flakes similar to pure MXene films. Additionally, (0002) peaks of (MXene/TAEA)$_n$ shifted from 6.89° for pure MXene films to 6.38°, which showed a uniform pillaring effect of TAEA in LbL process[38]. This corresponds to an increase of 1.0 Å in the average interlayer spacing from 12.8 Å up to 13.8 Å, which means an average interlayer distance of 1.38 nm between the MXene flakes in the multilayer films.

We note that this is in fact a very small increase in interlayer spacing even if we consider the small size of the TAEA molecule, which we believe results from complex interface interactions between MXene flakes and TAEA in a process which is yet not fully characterized and understood in this or other reported polymeric systems (e.g., MXene/PVA[20], MXene/CTAB[38], or MXene/PANI films[21]).

Additionally, the measurements of contact angles show that the TAEA layer is more hydrophilic than the MXene layer (Fig. 2f and Supplementary Fig. 10). The TAEA pillars within the pillared

architecture can, therefore, enhance the accessibility of the aqueous electrolytes into the MXene interlayers for efficient ion transfer and diffusion[1].

Figure 2d shows the electrical properties of $(MXene/TAEA)_n$ on silicon wafers as a function of n. The insulating substrate became a conductor after the assembly of a single (MXene/TAEA) bilayer. The sheet resistance of the LbL films then decreased sharply to $154\,\Omega\,sq^{-1}$ ($7.3 \times 10^4\,S\,m^{-1}$) for 20 bilayers. This conductivity is similar to that of films fabricated using small $Ti_3C_2T_x$ flakes ($7.78 \times 10^5\,S\,m^{-1}$)[39] and $Ti_3C_2T_x$ MXene clay ($1.5 \times 10^5\,S\,m^{-1}$)[16] which were made by similar methods to those used here, i.e., combined the etching by $LiF + HCl$ solution and delamination by sonication. This sheet resistance is notably lower than that of polymer-based LbL MXene films: $830\,\Omega\,sq^{-1}$ for $(MXene/PEI)_{20}$ (Supplementary Fig. 12), $8\,k\Omega\,sq^{-1}$ for $(MXene/PDAC)_{20}$[25], and $1400–450\,\Omega\,sq^{-1}$ for (MXene-PVA/CNT-PSS)$_n$[26], because polyelectrolytes form big insulating gaps between adjacent MXene flakes, such as in $(MXene/PEI)_n$ with an $8.71\,Å$ increase of interlayer spacing (Supplementary Fig. 9), and interrupt the electron transport between the flakes[32].

**LbL assembly of $(MXene/TAEA)_n$ on porous substrates.** To demonstrate the versatile fabrication of MXene multilayers over large areas on various types of substrates, we chose nonwovens and cellulose paper as examples of fibrous structures, and cellulose nanofiber (CNF) aerogels and melamine foams as examples of porous 3D frameworks (Supplementary Fig. 13) for LbL assembly of $(MXene/TAEA)_n$ using a spray-LbL self-assembly technique. Spray-LbL allows fast and uniform coating of porous and complex surfaces[40]. Previously, spray-LbL assembly of $TiO_2$ nanoparticles[35], multi-walled CNT[41], and polyelectrolytes[42] onto electrospun mats, multi-walled CNT onto carbon paper[43], and silica nanoparticles onto cotton fibers[40,44] have been demonstrated.

SEM images showed conformal coatings of $(MXene/TAEA)_n$ on all the substrates (Fig. 3 and Supplementary Fig. 14), and energy-dispersive X-ray spectroscopy (EDS) mappings of Ti confirmed a homogeneous disposition of $Ti_3C_2T_x$ MXene. The mass loading of $(MXene/TAEA)_n$ multilayers, for example on the nonwoven fibers, increases linearly with the bilayer numbers with an average increase of $5.54\,mg\,g^{-1}$ per bilayer (Supplementary Fig. 15). This rate is faster than those we observed on planar PET film because the porous structure of nonwovens has higher specific surface area to deposit more MXene flakes.

The highly conformal growth of $(MXene/TAEA)_n$ multilayers is mainly attributed to the spray-LbL process, which aerosolizes the dispersions of $Ti_3C_2T_x$ MXene flakes and the solution of

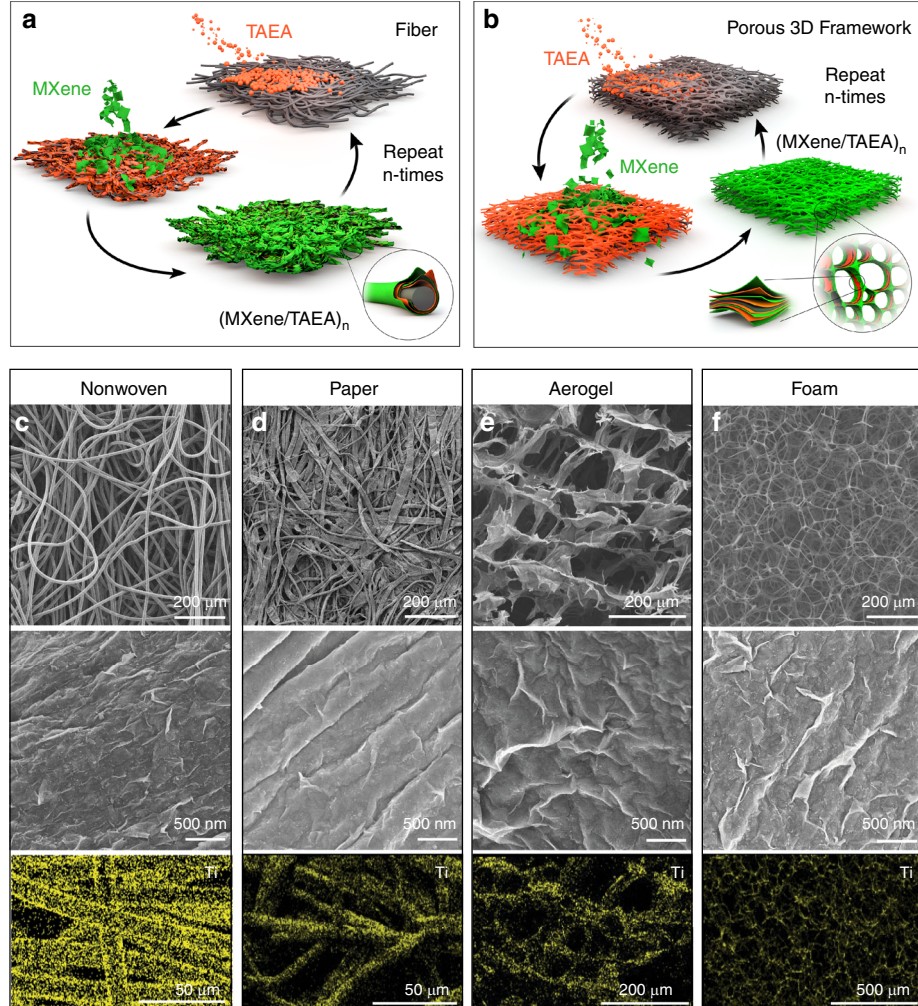

**Fig. 3** $(MXene/TAEA)_n$ multilayers on porous structures. Schematic illustration of the LbL self-assembly of $(MXene/TAEA)_n$ multilayer films onto **a** fibers and **b** foams. SEM images and EDS elemental mappings of the $(Ti_3C_2T_x/TAEA)_6$ multilayer films onto **c** nonwoven, **d** cellulose paper, **e** CNF aerogel and **f** melamine foam

TAEA and generates a strong convection and rearrangement between the interface of MXene flakes and TAEA[22,35]. We also provided a pressure gradient, using a vacuum suction on the back side of the substrates during the LbL self-assembly, to further force the aerosolized MXene or TAEA through the entire porous substrates (Supplementary Fig. 16).

**Electromechanical properties of (MXene/TAEA)$_n$ multilayers.** We explored the electromechanical response of (MXene/TAEA)$_n$ multilayers on the different substrates under various mechanical deformations. We first measured the changes of the sheet resistances of (MXene/TAEA)$_n$ on planar PET substrates as a function of the bending radius and cycle numbers. The results show a normalized resistance ($R/R_0$) of 1.9 at a bending radius of 1.5 mm (Fig. 4a) and retain a normalized resistance of 1.52 at final planar state after 1000 bending cycles (Fig. 4b). The (MXene/TAEA)$_n$ on PET connected in a circuit is able to light an LED during both bending and twisting (Inset Fig. 4b and Supplementary Movie 1). The sheet resistances of (MXene/TAEA)$_n$ on the porous substrates showed a similar trend to that of planar substrates, i.e., a sharp decrease in the resistance with increasing n (see nonwoven-based samples in Fig. 4c). The (MXene/TAEA)$_n$ on nonwovens were also resistant to extreme mechanical deformation (Supplementary Movie 2), for example, a knotted nonwoven showing a small normalized resistance of ~3.2 (Fig. 4c).

Foams and aerogels have higher surface area than planar and fiber-based substrates and offer the possibility of mechanical compression[42]. We further evaluated the effect of compressive strain on the electrical properties of (MXene/TAEA)$_n$ multilayers. Figure 4d shows that the normalized resistance of (MXene/TAEA)$_6$ on foams decreased almost linearly up to 50% compression. We could further compress the foam to 80% strain and return to the uncompressed state again retaining a

normalized resistance of 1.23. To the best of our knowledge, this is the only report of compressible 3D MXene-based materials (Supplementary Movie 3). The structures that we have developed here could be used in the design of fully 3D MXene-based energy storage devices[45].

**Electrochemical performance of (MXene/TAEA)$_n$ multilayers.** To evaluate the practical energy storage performance of the (MXene/TAEA)$_n$ multilayers, we fabricated symmetrical all-solid-state supercapacitors (Fig. 5a) using (MXene/TAEA)$_n$ on PET films as flexible electrodes and PVA/$H_2SO_4$ as solid-state electrolyte and separator. The devices with different bilayer numbers, n, all showed rectangle-shaped cyclic voltammetry (CV) curves (Fig. 5b) and symmetric triangle-shaped charge-discharge profiles (Fig. 5c), indicating an ideal capacitive behavior. The areal capacitances increased significantly with n (Fig. 5f), indicating that the LbL self-assembly enables precise control over the charge storage capability of MXene multilayers by simply altering the number of bilayers[46].

The CV curves (Fig. 5d) of the (MXene/TAEA)$_{20}$ displayed higher rate capability with an areal capacitance of 4.8 mF cm$^{-2}$ at a scan rate of 2 mV s$^{-1}$ and maintained a capacitance of 3.1 mF cm$^{-2}$ at a high scan rate of 200 mV s$^{-1}$ (Fig. 5f). The charge-discharge profiles of (MXene/TAEA)$_{20}$ have symmetric, linear shape at current densities ranging from 0.05 to 1 mA cm$^{-2}$ (Fig. 5e), and further demonstrated a higher rate-capability compared to samples with a fewer number of bilayers (Supplementary Fig. 17). We measured the highest volumetric capacitance of 583 F cm$^{-3}$ for (MXene/TAEA)$_6$ at a scan rate of 2 mV s$^{-1}$ (Fig. 5g). (MXene/TAEA)$_{20}$ electrodes had capacitance values of 534 F cm$^{-3}$ (165 F g$^{-1}$) at the scan rate of 2 mV s$^{-1}$ and 349 F cm$^{-3}$ (108 F g$^{-1}$) at 200 mV s$^{-1}$, exhibiting the highest capacitance retention of 65.5% at higher rates compared to the

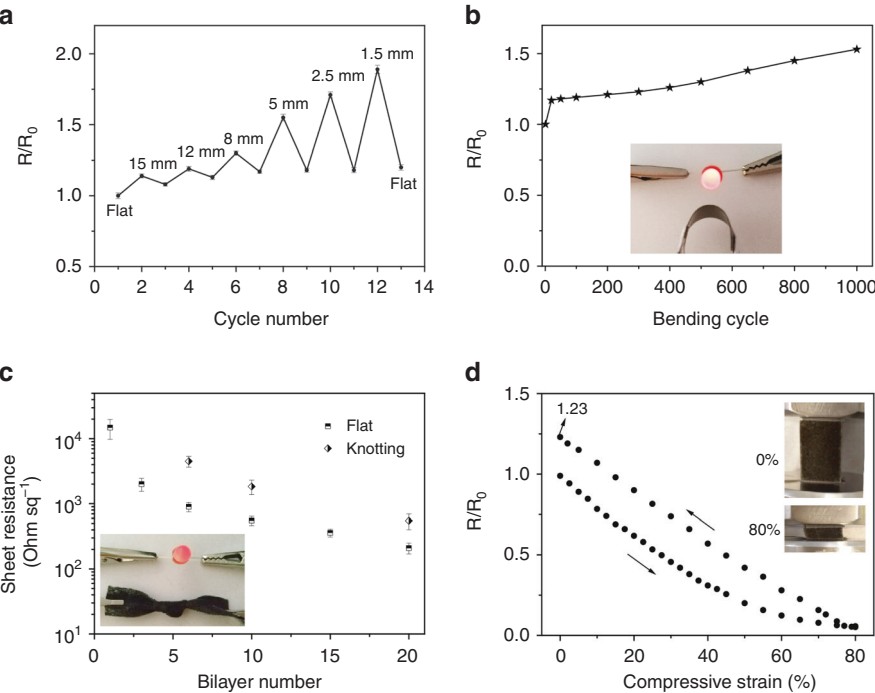

**Fig. 4** Electromechanical properties of (MXene/TAEA)$_n$ multilayers. **a** Normalized resistance $R/R_0$ ($R_0 = 1.63$ kΩ) as a function of bending radius and **b** cyclic stability with a bending radius of 5 mm of (MXene/TAEA)$_{10}$ on PET sheet. Inset in b shows a digital photograph of (MXene/TAEA)$_{20}$ on PET sheet under bending condition with a LED connection. **c** Sheet resistances of (MXene/TAEA)$_n$ on nonwovens at flat and the knotted conditions. Inset shows a digital photograph of knotted (MXene/TAEA)$_{30}$ on nonwoven with a LED connection. **d** Normalized resistance $R/R_0$ ($R_0 = 36.2$ kΩ) as a function of compressive strain for the (MXene/TAEA)$_6$ on melamine foams, and the inset shows a photograph of the compression/resistance measurement for (MXene/TAEA)$_n$ on foams in uncompressed and with 80% compressive strain

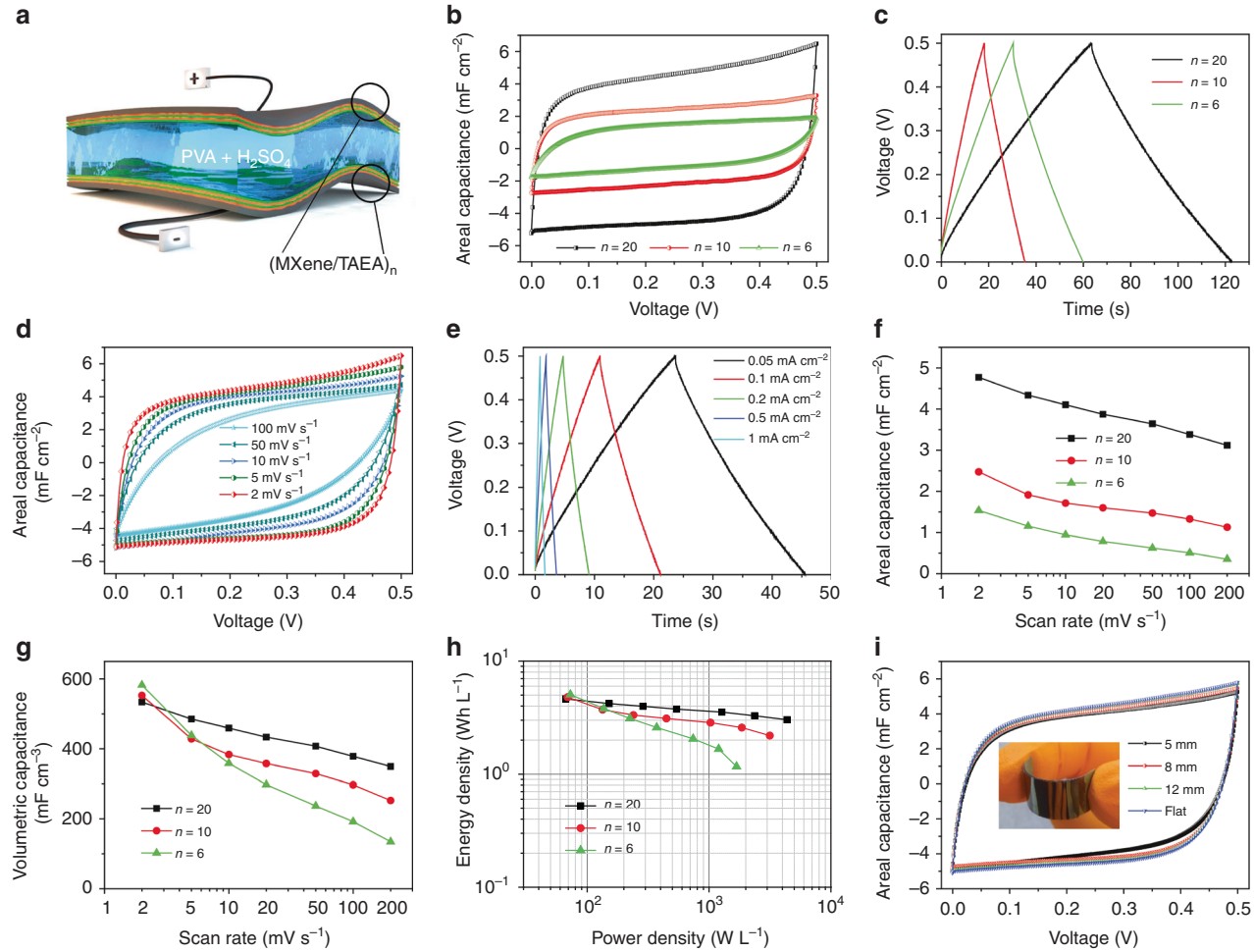

**Fig. 5** Electrochemical performance of flexible solid-state supercapacitors based on (MXene/TAEA)$_n$ electrodes. **a** Schematic illustration of a solid-state supercapacitor based on (MXene/TAEA)$_n$ on PET sheet as electrodes and PVA/H$_2$SO$_4$ as the electrolyte. **b** CV curves at a scan rate of 2 mV s$^{-1}$, and **c** charge-discharge profiles at a current density of 0.02 mA cm$^{-2}$ for (MXene/TAEA)$_n$ with bilayer number n. **d** CV curves at different scan rates and **e** charge-discharge profiles at different current densities of (MXene/TAEA)$_{20}$. **f** Areal capacitances, and **g** volumetric capacitances of (MXene/TAEA)$_n$ with different bilayer numbers, n, at different scan rates. **h** Ragone plot of (MXene/TAEA)$_n$ solid-state supercapacitors based on total active electrode volume. **i** CV curves of (MXene/TAEA)$_{20}$ solid-state supercapacitors measured for different degrees of bending, and the inset shows a photograph of the solid-state supercapacitor with symmetrical electrodes on PET sheet

electrodes with fewer bilayers (Supplementary Fig. 18). Note that the (MXene/TAEA)$_{20}$ showed a higher specific capacitance compared with (MXene/PEI)$_{20}$ (Supplementary Fig. 19), because the smaller gaps formed between adjacent MXene flakes as the result of pillaring by TAEA molecules generated a higher packing density of electrochemically active materials and a smaller equal series resistance.

We used the Ragone plot shown in Fig. 5h to evaluate the energy and power densities of the (MXene/TAEA)$_n$-based flexible all-solid-state supercapacitors based on the total electrochemically active electrode volumes. Our devices achieved the highest volumetric energy density of 5.1 Wh L$^{-1}$ at a power density of 72.8 W L$^{-1}$, and could maintain an energy density of 3.0 Wh L$^{-1}$ at a high power density of 4400 W L$^{-1}$. These volumetric energy densities are higher than previously reported flexible all-solid-state supercapacitors based on MXene/graphene[28], graphene hydrogel[47], graphene/polyaniline[48], and carbon nanotube[49] (Supplementary Fig. 20). Additionally, the solid-state supercapacitors based on (MXene/TAEA)$_{20}$ exhibited an excellent cycling stability with 90.3% capacitance retention after 10,000 cycles (Supplementary Fig. 21). The fabricated devices were

flexible and showed an almost steady capacitance response at different bending radius. For example, we observed a less than 10% capacitance change at a small bending radius of 5 mm (Fig. 5i). We also evaluated the electrochemical performances of (MXene/TAEA)$_n$ on porous substrates as electrodes in the all-solid-state supercapacitor setup (Supplementary Figs. 22, 23). Notably, (MXene/TAEA)$_6$ on aerogels showed an areal capacitance of 48 mF cm$^{-2}$. This originated from the 3D porous framework which provides a high internal surface area to load MXene flakes at a certain footprint area[34].

The excellent electrochemical performance of our devices result from the LbL pillared multilayer structures, which we believe have three unique features: (i) The pillars of hydrophilic TAEA between the interlayer spacing of MXene flakes facilitates proton access to more shallow- and deep-trap sites inside MXene interlayers, enhancing the intrinsic pseudocapacitive response of the electrode[16,50]. (ii) The expanded interlayer spacing improves ion movement with a small charge transfer resistance ($R_{ct}$) of 2.3 Ω obtained from electrochemical impedance spectroscopy (EIS) (Supplementary Fig. 17f), and provides more ion-accessible surface for redox reactions and interlayer volume to

accommodate more electrolyte ions for charge storage[21]. (iii) The assembly of MXene flakes with face-to-face quasi-intimate interface contact leads to an ordered dense packing which results in high volumetric capacitances and also generates higher electronic conductivity for fast charge transport at higher rates.

## Discussion

We have demonstrated an efficient strategy for fabricating pillared MXene multilayers using LbL self-assembly from water. These films have five noteworthy features: (i) They grow linearly for each assembled bilayer, which indicates our LbL self-assembly is a highly precise method. (ii) They have an ordered structure similar with the pristine MXene films, as shown by XRD data, but with an increased interlayer spacing of 0.1 nm. The formation of 2D ordered structures with LbL self-assembly, and this level of precision in pillaring is, to the authors' knowledge, distinct from previous literature reports. (iii) Their conductivity approaches $10^5 \, S \, m^{-1}$ which is similar to that of a pristine MXene film, because the tunneling distance between the pillared MXenes in multilayers is very small. (iv) They can act as electrodes in solid-state supercapacitors and deliver a high energy density of 3.0 Wh $L^{-1}$ at a high power density of 4400 W $L^{-1}$. These values are higher than most previously reported carbonaceous electrode-based all-solid-state supercapacitors. These results stem, firstly, from the enhanced interlayer spacing which facilitates proton access to deep-trap sites inside MXene interlayers, and, secondly, from the dense packing of single flakes in multilayers which increases volumetric capacitance without compromising electronic conductivity during fast charge/discharge. v) They can be fabricated over large surface areas and various substrates such as nonwovens and even aerogels. When assembled onto these substrates, they exhibit the resistance to extreme deformations such as bending, twisting and knotting and even extreme compression (up to 80% strain). We attribute this to the pillaring effect of TAEA which reinforces the interconnection between MXene flakes.

This self-assembly strategy should easily be applicable to other 2D colloidal building blocks such as graphene, layered transition metal oxides, and transition metal dichalcogenides. We believe that the combination of several 2D materials with this LbL method could also open the path towards self-assembled 2D heterostructures for energy storage[45], or electronic devices[1].

## Methods

**Materials.** Tris(2-aminoethyl)amine (TAEA), branched PEI (60 kDa), poly(vinyl alcohol) (PVA) (Mw 89,000–98,000, 99+% hydrolyzed), sulfuric acid ($H_2SO_4$) (≥97.5%), and the polyethylene terephthalate (PET) film were purchased from Sigma Aldrich. The anionic potassium polyvinyl sulfate (KVPS) was obtained from Wako Pure Chemicals, Osaka. A commercial nonwoven was obtained from ICA Sweden, and a Chinese traditional cellulose paper (XuanZhi) from Alibaba and melamine foams (MF) from Recticel. CNF aerogel was prepared as reported previously[34,45].

**Synthesis of $Ti_3C_2T_x$ MXene.** Delaminated-$Ti_3C_2T_x$ (d-$Ti_3C_2T_x$) MXene solution was prepared according to previous reports in the literature[17]. In a typical synthesis, 2 g of LiF powder was dissolved in 40 mL 6 M HCl solution. The solution was stirred for 5 min to LiF completely dissolve in the acidic solution. Then 2 g of $Ti_3AlC_2$ powder was slowly added to the etchant mixture (over 10 min). An ice bath was used to avoid excessive heat generation while adding the MAX phase powder to the etchant. The etching was carried out at 35 °C for 24 h under continuous stirring at 550 rpm. After etching was complete, the exfoliated powders were washed with DI water and centrifuged several times until the pH was about 6. The final MXene powders were dispersed in deaerated DI Water (in a 1 g to 100 mL ratio) and were probe sonicated in an ice bath for 1 h (35% power). The obtained solution was again centrifuged at 3500 rpm for 1 h and the supernatant was collected (referred to as $Ti_3C_2T_x$ MXene solution). The concentration of the MXene solutions was measured by filtering a known volume of the solution over Celgard® membrane and weighing the obtained freestanding MXene film after it was completely dried.

**Spin-assisted immersive-LbL assembly of MXene.** We used silicon wafers and PET films as the planar substrates for the spin-assisted immersive-LbL assembly of MXene. Prior to the spin-assisted immersive-LbL assembly of MXene, silicon wafer and polyethylene terephthalate (PET) were cut into desired strips and were treated with oxygen plasma (Optrel GBR, Multi-stop) for 10 min at a high level under vacuum. In the process of spin-assisted immersive-LbL assembly, we used a dipping robot (nanoStrata Inc.) with a spinning model, and the $Ti_3C_2T_x$ MXene, PEI, TAEA, spermidine and tris(3-aminopropyl)amine (TAPA) solutions with a concentration of 1 g $L^{-1}$. The treated substrates were first dipped into TAEA solution for 5 min and were then rinsed 3 times by Milli-Q water for 3, 2, 1 min per time to remove the weakly absorbed molecules. After that, the cation-coated substrates were dipped into the $Ti_3C_2T_x$ MXene dispersion for 5 min and then rinsed with Milli-Q water as the same steps above. This cycle made one bilayer of (MXene/TAEA)$_1$, and the bilayer was repeated to fabricate the desired multilayers, denoted as (MXene/TAEA)$_n$ where n is the bilayer number. The as-prepared multilayer films were dried at room temperature under a vacuum condition. The multilayers over 20 bilayers were dried twice, for example, (MXene/TAEA)$_{30}$ was dried first after 20th bilayer and last after 30th bilayer.

**Spray-LbL assembly of MXene.** We used a vacuum-assisted spray-LbL assembly process to coat the MXene multilayer films onto the different non-metal porous substrates. In the process, the $Ti_3C_2T_x$ MXene and TAEA solutions were used with a concentration of 1 g $L^{-1}$. All porous substrates used were cut into the desired dimensions and were laid on a cellulose membrane which had been fixed in an adjustable-flow vacuum system. Before the deposition of (MXene/TAEA)$_n$ bilayers, the vacuum-assisted system was opened to hold the substrates by vacuum. The airbrushes (FA-180A, Nozzle: 0.20 mm) were used to spray the atomized solutions which were driven by the compressed ultrapure Ar regulated to 20 psi and were held at a sufficient distance to reach the entire surface of substrates simultaneously. The substrates were first sprayed with TAEA solution for 3 s and were then rinsed by spraying Milli-Q water for 5 s to remove the weakly absorbed TAEA molecule. Subsequently, the TAEA-coated substrates were sprayed by the MXene for 3 s and then rinsed by spraying Milli-Q water. The cycle was repeated to fabricate the (MXene/TAEA)$_n$ films with the desired thickness.

To further show the LbL assembly of MXene onto the larger surface of 3D CNF aerogel and melamine foam, we used our previously reported rapid-LbL assembly method[1]. Briefly, the $Ti_3C_2T_x$ MXene and TAEA solutions were poured sequentially on the top of the aerogel or foam and then were forced through by applying a vacuum pressure on their bottom. The samples were rinsed with Milli-Q water after each step. The cycle was repeated to fabricate the (MXene/TAEA)$_n$ films with the desired bilayer numbers.

**Material characterization.** SEM images and EDS spectra were taken by a high-vacuum field emission scanning electron microscope with (FE-SEM, Hitachi S-4800, Hitachi Corp., Japan). AFM images were captured by a Multimode 8 Atomic Force Microscope (AFM) with a NanoScope V controller (Bruker Corp., USA) in the ScanAsyst® mode. The contact angles were measured on contact angle meter with CAM200 model (KSV Instruments LTD). QCM-D (E4, Q-Sense AB, Västra Frölunda, Sweden) was used to investigate the formation of (MXene/TAEA)$_n$ multilayers. XRD spectra were obtained from the MXene multilayers assembled on silicon wafers and were conducted on a PANalytical X'Pert PRO powder diffractometer in the range of 4–30° (2θ). The interlayer spacing $d$ (nm) of MXene multilayers was calculated according to the following equation (1):

$$d = \frac{\lambda}{2sin\theta_{0002}} \quad (1)$$

where $\lambda$ ($\lambda = 0.15406$ nm) is the wavelength of X-ray used, and $\theta_{0002}$ is the scattering angles of the (0002) peak of MXene multilayers.

The thickness of MXene multilayers was obtained from the cross-sectional SEM images, by averaging the values from 10 different sites. The mass loading of MXene multilayers was obtained from 5 parallel samples, using a balance (±0.01 mg) to weigh the mass change in a given area before and after the assembly of (MXene/TAEA)$_n$ multilayers.

The zeta potential of $Ti_3C_2T_x$ MXene dispersions at different pH was tested using a Zetasizer ZEN3600 (Malvern Instruments Ltd., U.K.). The charge density of 1 g $L^{-1}$ TAEA at different pH was titrated using a 716 DMS Titrino, Metrohm, with a standard chemical of KVPS (a charge density of $-0.379$ μeg $mL^{-1}$ at 0.05 g/L) as the titrant and ortho-toluidine blue as the indicator. The charge density $Q$ (μeg $mL^{-1}$) of TAEA was calculated according to the following equation (2):

$$Q = \frac{0.379 \times V_{KVPS}}{V_{TAEA}} \quad (2)$$

where $V_{KVPS}$ is the volume of KVPS solution and $V_{TAEA}$ is the volume of TAEA solution.

The sheet resistances of MXene multilayers were measured by SourceMeter 2401 KEITHLEY, Beaverton through a 2-point probe technique. The samples were cut into defined dimensions of 1.5 cm length and 1 cm wide, and the short edges were coated with silver paste to avoid the contact resistance between the film and metal probe. The sheet resistance $R_s$ ($\Omega \, sq^{-1}$) and conductivity $\sigma$ (S $m^{-1}$) of

MXene multilayers was calculated according to the following equations:

$$R_s = R\frac{W}{L} \tag{3}$$

$$\sigma = \frac{1}{R_s \times d} \tag{4}$$

where $R$ is the resistance measured, $W$ and $L$ are the width and length of the real area measured, and $d$ is the thickness of MXene multilayers.

Electromechanical properties of MXene multilayers on melamine foams was measured by placing the samples with flat surfaces against two plates of aluminum foils placed inside of an Instron 5594 universal testing machine (Instron Corporation, High Wycombe, UK), and connecting the foils to a multimeter to monitor the resistance change with the compressive strain. The back sides of aluminum foils were protected by insulating plastic sheets, and a velocity of 10% $min^{-1}$ in compression and extension was chosen.

**Synthesis of PVA/$H_2SO_4$ electrolyte**. 1 g of PVA was added in 10 mL of Milli-Q water. The whole mixture was then heated up to 85 °C under stirring until the solution turned clear, and then cooled under ambient conditions. Subsequently, 3 g of concentrated $H_2SO_4$ was added to the above solution and then stirred vigorously for 1 h at room temperature.

**Fabrication of (MXene/TAEA)$_n$-based electrodes**. We fabricated the electrodes of the (MXene/TAEA)$_n$ on PET through tailoring the films to the desired shape (1 cm × 2 cm) and coating one end with the silver paste to decrease the contact resistance. The electrodes of (MXene/TAEA)$_n$ on nonwoven, and CNF aerogels were prepared as follows: The composites were first tailored to the desired shape, and then one end was connected with a strip of nickel foil using the silver paste. The contact point was protected using paraffin wax to prevent the electrolyte from the contact point.

**Fabrication of solid-state supercapacitors**. To assemble the solid-state devices, the PVA/$H_2SO_4$ gel electrolyte was slowly poured on the working area of the ($Ti_3C_2T_x$/TAEA)$_n$ electrodes, and then they were dried under vacuum condition for overnight to vaporize the excess water and to remove the trapped oxygen from the electrolyte. After that, the two identical electrodes were pressed face-to-face together, leading to a structure in which the solid electrolyte on each electrode constructed a thin separating film.

**Electrochemical characterization**. We used a VMP3 potentiostat (Biologic, France) to conduct all electrochemical characterizations. We measured the EIS spectra over a frequency range of 10 mHz to 200 kHz with a perturbation amplitude of 10 mV.

In the three-electrode configuration, (MXene/TAEA)$_n$ multilayers were used as the working electrode, and a Pt foil and an Ag/AgCl electrode were used as the counter electrode and the reference electrode, respectively. A 1 M $H_2SO_4$ solution was used as the electrolyte. The areal capacitance $C_A$ (mF $cm^{-2}$) was calculated from the CV curves according to the following equation:

$$C_A = \frac{1}{A_s \Delta V v} \int_{V_i}^{V_v} i(V) dV \tag{5}$$

where $A_s$ is the effective area of working electrodes, $v$ is the scan rate, $\Delta V$ is voltage window, $i(V)$ is the current, $V_i$ is the initial potential and $V_v$ is the vertex potential.

In the solid-state supercapacitors, the areal capacitance $C_A$ (mF $cm^{-2}$) per electrode was calculated from the CV curves according to the following equation:

$$C_A = \frac{4}{A \Delta V v} \int_{V_i}^{V_v} i(V) dV \tag{6}$$

where $A$ is the effective geometric area of the two electrodes.

Gravimetric capacitance $C_s$ (F $g^{-1}$) and $C_v$ (F $cm^{-3}$) per electrode were calculated according to the following equation:

$$C_s = C_A/S \tag{7}$$

$$C_V = C_A/d \tag{8}$$

where $S$ is the mass loading of (MXene/TAEA)$_n$ multilayers and $d$ is the thickness of (MXene/TAEA)$_n$ multilayers.

We also calculated the $C_A$ (mF $cm^{-2}$) per electrode from charge-discharge profiles according to the following equation:

$$C_A = \frac{4i\Delta t}{A \Delta V_d} \tag{9}$$

where $i$ is the current, $\Delta t$ is the discharge time, $\Delta V_d$ is the discharge voltage window after the $IR$ drop.

Volumetric energy density $E_v$ (Wh $L^{-1}$), and power density $P_v$ (W $L^{-1}$), based on two electrodes, was calculated according to the following equations:

$$E_V = \frac{C_V(\Delta V)^2}{8 \times 3.6} \tag{10}$$

$$P_V = \frac{E_V \times 3600}{\Delta t} \tag{11}$$

## Data availability

The data that support the findings of this study are available from the corresponding authors upon a reasonable request.

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

## Acknowledgements

M. H. acknowledge the European Research Council (ERC Starting Grant 715268) for funding. W. T., Z. W., and M. H. acknowledge the Wallenberg Wood Science Centre at KTH for funding through the Knut and Alice Wallenberg Foundation. Z. W. acknowledges the Wenner-Gren Foundation for funding. A.V.M. acknowledges the support from Alabama EPSCoR Graduate Research Scholar Program (GRSP Round 12 and 13). M.B. acknowledges the support from Auburn University's Inramural Grants Program (IGP). We thank Innventia (RISE Sweden) for providing nanocellulose.

## Author contributions

W.T. designed, prepared and characterized the LbL MXene multilayers, conducted the electrochemical test, analyzed the data and wrote the manuscript with supervision by M. H. A.V.M. prepared and characterized MXene dispersions, and contributed to writing the manuscript with supervision by M.B. A.V.M. designed the schematic illustrations. Z. W. designed and analyzed the LbL assembly process. M.B., A.V.M., and O. L. analyzed the electrochemical data. M. H. contributed to the development of ideas, experimental design, and writing. All the authors contributed to the analysis of the data and writing of the manuscript.

## Additional information

**Competing interests:** The authors declare no competing interests.

