## [Peer Review File · Nature Communications]

Reviewers' comments:

Reviewer #1 (Remarks to the Author):

In this manuscript, the authors used a layer-by-layer (LbL) self-assembly of pillared two-dimensional multilayers on substrate. The material used is MXene/TAEA. It is claimed that TAEA of the LBL expands the interlayer spacing of MXene layers by only 1.03 Å thereby reinforcing the interconnection between MXene flakes. It is also proposed that TAEA-pillared MXene multilayers are resistant to mechanical deformation. They assembled a supercapacitor and demonstrated its performance. My questions/comments are:

1. When a positively charged small molecule, TAEA, is introduced as the interlayer pillaring component for the LbL self-assembly how the electronic conductivity between the MXene layers is improved? In fact, this is going to affect the charge transport.
 2. How a positively charged TAEA improves the capacitance? It is clear that the interlayer distance is increased, but the positively charged TAEA repels the cation intercalation.
 3. The electrical conductivity values should be consistent everywhere. In the same paragraph in page 8, the authors compare the conductivity with sheet resistance. This is confusing to the readers. What is the electrical conductivity of TAEA? How TAEA incorporation affects the conductivity.
 4. The areal capacity reported is extremely small, 4.8 mF cm⁻² at a scan rate of 2 mV s⁻¹.
 5. It is mentioned, "cycling stability with 89% capacitance retention after 5,000 cycles at a current density of 0.1 mA cm⁻²" is an excellent value. For a supercapacitor this is a below average value!
 6. The discussion of the results and the reasons for enhanced performance is very weak. It is mentioned that the interlayer spacing is increased without giving any surface area values. Also, EIS of electrodes with and without TAEA is not compared. This is important to understand the mechanism of charge transfer in a pseudocapacitive environment.
- In summary, the authors developed a multilayer MXene and TAEA and claim that it is an excellent material for supercapacitors when the actual electrochemical performances are below average. I do not recommend the publication of this paper in Nature Communications.

Reviewer #2 (Remarks to the Author):

The authors report on a new layered by layered stacking of 2D sheets of MXene using TAEA to control the spacing between MXene sheets and improve their bonding. Further, they showed the possibility of making all-solid-state supercapacitors with high volumetric capacitance and its mechanical stability. The work is novel at the level of Nature Comm. and 2D community and MXene community, in particular, can benefit from this work. However, a major revision is needed:

1. The bilayer stacking, interlayer spacing, and a MXene layer thickness need clear definitions. For example in "based on the thickness of single MXene flake (~ 2.5 nm), and the interlayer spacing (~1.4 nm) and the bilayer thickness (4.5 nm) of LbL MXene films" the interlayer spacing from XRD includes the thickness of a MXene flake (~1nm), so the gallery between the layers is only 0.4 nm. The bilayer thickness needs to be recalculated.
2. In "The expansion of the interlayer spacing indicates that TAEA molecules have intercalated into the adjacent MXene flakes ..." It is not clear how the LbL would cause intercalation. It is more of stacking molecules over MXene by separate dip coating (LbL) than intercalation.
3. Since TAEA and MXene are oppositely charged, have authors considered mixing the two solutions to create self-assembly formation of this hybrid structure. If it is not possible, a clear explanation is needed.
4. Authors say "ordered structures in LbL MXene films arise only when the counter phase is the small-molecule TAEA" What is the limit on this "small" molecule size (in nm)? More specific

details.

5. More details on the AFM and XRD of MXene TAEA is needed. Authors need to explain why there is no height change after TAEA in AFM and then there is a slight increase in the stacking of the flakes in XRD. A discussion about the exact size of the TAEA is needed.

6. In "small gap between the interface contact of MXene flakes resulting in high electronic conductivity," It is not clear why the gap between MXene flakes helps the conductivity.

7. In "This conductivity is close to that of pure Ti₃C₂T_x MXene" the conductivity values are certainly lower than the values for MXene papers. More recent refs are needed. However, authors can mention their conductivity is similar to the MXene made in similar conditions (sonications, flake size, etc.).

8. Authors mention: "This value is notably lower than that of polymer-based LbL MXene films: 830 Ω sq⁻¹ for (MXene/PEI)₂₀" relating it to the big size of molecules. To be more specific, XRD of MXene/PEI is needed to show the effect of the large PEI molecules.

9. Why does the MXene/TAEA planar flat in fig 4a has a curvature? Was there any plasticity?

10. In the electrochemical section "The expanded interlayer spacing accelerates ion movement with a small charge transfer resistance". However, the increase in the interlayer spacing is not significant (~0.4 Å). More explanation is needed here and comparison with previously reported interlayer spacing of MXene.

Minor issues:

Full detail of the sequential LbL is lacking.

Can authors make the zeta potential axis normal (increasing in the y-axis)

Figure 2f. It is not clear why the Si substrate changes color from dark to bright gray by increasing the #n, from left to right.

The term "bilayer" is not very clear. Does it mean a layer of MXene with TAEA or 2 layers of MXene with TAEA?

The term "overlapped face-to-face" is not clear. Does this mean stacking of the flakes?

Reviewer #3 (Remarks to the Author):

In this manuscript, a positively charged small molecule, tris (2-aminoethyl) amine (TAEA) was introduced as the interlayer pillaring component for the LbL self-assembly of Ti₃C₂T_x MXene, and pillared multilayers of (MXene/TAEA)_n was fabricated. The pillared multilayers of (MXene/TAEA)_n was used to solve the problem of highly conducting performance of multilayered structure MXene or other 2D materials with single flakes precision in each layer, and with a pillaring between the 2D layers. The materials grow linearly with 4.5 nm for each assembled bilayer, which indicates that there have 1-2 single flakes in each layer. In these LbL architectures, MXene flakes are assembled in a face to-face quasi-intimate contact leading to a high packing density. The intercalated pillars of TAEA further enable a small gap between the interface contact of MXene flakes resulting in high electronic conductivity, and in a slightly expanded interlayer spacing between the individual MXene flakes which accelerates ion diffusion and provides a facile access to the titanium atoms on the surface of MXene layers for fast pseudocapacitive charge storage. The multilayer conductivity approaches 105 S m⁻¹ which is similar to that of a pure MXene film. The multilayer can act as electrodes in a solid-state supercapacitor and display a high energy-density of 3.0 Wh L⁻¹ at a high power-density of 4400 W L⁻¹. The multilayer films are conformal, conductive, and resilient to bending and compression, and the technique of (MXene/TAEA)_n films was further demonstrated to expand onto various non-metallic substrates including planar films, nonwovens, paper, foams, and even aerogels.

The topic is very interesting. The paper is well written, organized, the figures number is fine. However, before publication in Nature Communication as free open access paper, a careful check of English language should be made in the interest of the authors, who diversely could lost the right interest from a broad readers. Therefore I strongly suggest to revise the English language.

- 1、 In Page 7, Line 158-161 and in the figure 2b, "This is the most remarkable result of this work since it suggests that each bilayer contains only 1-2 single MXene flakes, based on the thickness of single MXene flake (~ 2.5 nm), and the interlayer spacing (~1.4 nm) and the bilayer thickness (4.5 nm) of LbL MXene films." In this sentence, the thickness of bilayer(4.5nm) was from the average with multiple layers of thickness, which is not according with the signal flake
If the compositional uniformity and thickness flatness of the intermediate portion and the edge of this multilayer structure could be described in detail, it would be easier to explain the distance between the layers and explain the hydrophobicity characteristics.
- 2、 What is the relationship between alternate completeness of two kind of layer materials and the capacitive properties of the pillared multilayers of (MXene/TAEA) n?
- 3、 In Figure 7 (sup), the difference in the contact angle among n3.5 and n10, n30 is not very obvious, does it mean that the two kind of layer materials are not completely alternated in the pillared multilayers of (MXene/TAEA) n?

We thank the Reviewers for their insightful and constructive comments. We have considerably revised the manuscript and the Supplementary Information with new data and analysis to address the comments made by the Reviewers. All updates in the manuscript and the Supplementary Information are highlighted in yellow. Please find below our point-by-point responses to Reviewers' comments, in which the original Reviewer comments are shown in blue text and our responses are in black. When we refer to the page, figure and reference numbers here, they are corresponding to ones in the revised manuscript and Supplementary Information.

Reviewer 1

In this manuscript, the authors used a layer-by-layer (LbL) self-assembly of pillared two-dimensional multilayers on substrate. The material used is MXene/TAEA. It is claimed that TAEA of the LBL expands the interlayer spacing of MXene layers by only 1.03 Å thereby reinforcing the interconnection between MXene flakes. It is also proposed that TAEA-pillared MXene multilayers are resistant to mechanical deformation. They assembled a supercapacitor and demonstrated its performance. My questions/comments are:

I-1. When a positively charged small molecule, TAEA, is introduced as the interlayer pillaring component for the LbL self-assembly how the electronic conductivity between the MXene layers is improved? In fact, this is going to affect the charge transport.

Response: We thank the reviewer for their comments. We should clarify that in our manuscript, we do not claim the electronic conductivity is enhanced in the LbL self-assembled (MXene/TAEA)_n films compared to the pure Ti₃C₂T_x MXene films. What we claim is that using small molecules in the LbL process does not adversely affect the conductivity of the self-assembled multilayer films and therefore, they have a comparable conductivity with pure MXene films fabricated with MXene flakes having similar lateral dimensions. Moreover, the (MXene/TAEA)_n multilayers show the highest electronic conductivity compared to all previously reported MXene multilayers fabricated by LbL self-assembly using conventional polymers as counter phases (such as polyethyleneimine (PEI) [ref 27], or poly(diallyldimethylammonium chloride) (PDAC) [ref. 25], or even with poly(sodium 4-styrene sulfonate) (PSS)-modified CNT [ref. 26]). The reason is that a small

molecule only leads to a small increase in the interlayer spacing of the multilayers and does not form a thick insulating layer between the MXene flakes, which most conventional polymers used in LbL process do. To clarify this point, we have revised the text on [page 2] in the revised manuscript:

“The TAEA-pillared MXene multilayers show the highest electronic conductivity of $7.3 \times 10^4 \text{ S m}^{-1}$ compared with all reported MXene multilayers fabricated by LbL self-assembly technique...”

We elaborate more on this topic in response 1-3.

1-2. How a positively charged TAEA improves the capacitance? It is clear that the interlayer distance is increased, but the positively charged TAEA repels the cation intercalation.

Response: When the TAEA is LbL self-assembled into the multilayers, the positive charges are balanced by the negative charges from the surface of MXene flakes through electrostatic adsorption (this is the principle of LbL process). Please note that the charge storage mechanism in $\text{Ti}_3\text{C}_2\text{T}_x$ MXene is based on intercalation pseudocapacitance through fast and reversible redox reactions, which in the H_2SO_4 electrolyte are carried out by intercalation of protons. When the charge of TAEA layers is balanced by MXene layers, the molecules cannot repel protons. Therefore, the presence of TAEA layers between MXene layers does not hinder the protons' intercalation from the acidic electrolyte and will further not hinder the redox reactions required for charge storage. Furthermore, the increase in the interlayer spacing (although it is small) will improve the ion (protons in this case) diffusion in the LbL MXene multilayer films resulting in their enhanced electrochemical performance.

To further clarify the effects and advantages of using TAEA, we compared the electrochemical performance of $(\text{MXene}/\text{TAEA})_{20}$ with those of polymer-based multilayer of $(\text{MXene}/\text{PEI})_{20}$, as shown below. We have added these CV and EIS data to Supplementary Fig. 19 in the Supplementary Information.

“Supplementary Fig. 19 | Electrochemical performances of (MXene/PEI)₂₀-based solid-state supercapacitors. (a) CV curves of (MXene/PEI)₂₀ at different scan rates compared with those of (MXene/TAEA)₂₀. (b) Charge-discharge profile of (MXene/PEI)₂₀ with a large IR drop. (c) Nyquist plots of (MXene/PEI)₂₀-based solid-state supercapacitors.”

The results show that the specific capacitance of (MXene/TAEA)_n multilayers was significantly higher than that of (MXene/PEI)_n at the same bilayer number, and we can control the capacitance by adjusting the bilayer number *n* [page 10-11 of the revised manuscript]. We have also added the following text on [page 11] in the revised manuscript to highlight this point:

“Note that the (MXene/TAEA)₂₀ showed a higher specific capacitance compared with (MXene/PEI)₂₀ (Supplementary Fig. 19), because the smaller gaps that were pillared by TAEA molecules between adjacent MXene flakes generated a higher packing density of electrochemically active material and a smaller equal series resistance.”

1-3. The electrical conductivity values should be consistent everywhere. In the same paragraph in page 8, the authors compare the conductivity with sheet resistance. This is confusing to the readers. What is the electrical conductivity of TAEA? How TAEA incorporation affects the conductivity?

Response: We had clarified the methods used for calculation of both sheet resistance and conductivity of the planar multilayer films and stated the corresponding equations and their correlations (in Eq. (3) and Eq. (4)) in the Method section in the manuscript. The reason that we compare sheet resistance values is that some of the cited references have not provided

conductivity values and have only reported sheet resistance values. Additionally, the conductivity and sheet resistance are measured from the entire composite multilayers, and not from any one of building blocks, such as MXene, or TAEA. To clarify the influence of TAEA on the conductivity, we have compared the sheet resistances of $(\text{MXene/TAEA})_n$ with that of $(\text{MXene/PEI})_n$ (Supplementary Fig 12) and all previously reported MXene multilayers fabricated by LbL self-assembly in the literature, showing that $(\text{MXene/TAEA})_n$ has the highest electronic conductivity. To further explain this point, we have also added the XRD patterns of MXene/PEI multilayers as Supplementary Fig 9 in the Supplementary Information:

“Supplementary Fig. 9 | XRD patterns of $(\text{MXene/PEI})_n$ compared with MXene film and $(\text{MXene/TAEA})_n$, and digital photographs of the $(\text{MXene/TAEA})_{20}$ and $(\text{MXene/PEI})_{20}$. The position of (0002) peaks of $(\text{MXene/PEI})_{20}$ shifts from 6.89° of pure MXene films to 4.10° , corresponding to an increase of 8.71 \AA in the interlayer spacing from 12.81 \AA up to 21.53 \AA .”

We have also revised the text on [page 8] in the revised manuscript:

“This sheet resistance is notably lower than that of polymer-based LbL MXene films: $830 \text{ } \Omega \text{ sq}^{-1}$ for $(\text{MXene/PEI})_{20}$ (Supplementary Fig. 12), $8 \text{ k } \Omega \text{ sq}^{-1}$ for $(\text{MXene/PDAC})_{20}$,²⁵ and $1400\text{-}450 \text{ } \Omega \text{ sq}^{-1}$ for $(\text{MXene-PVA/CNT-PSS})_n$,²⁶ because polyelectrolytes form big insulating gaps between adjacent MXene flakes, such as in

(MXene/PEI)_n with an 8.71 Å increase of interlayer spacing (Supplementary Fig. 9), and interrupt the electron transport between the flakes.^{32,}

I-4. The areal capacity reported is extremely small, 4.8 mF cm⁻² at a scan rate of 2 mV s⁻¹.

Response: Please note that the areal capacitances of the electrodes depend on the thickness of the electrode (i.e. mass loading). This can be seen in Fig. 5, where we show how the areal capacitance is increasing with increasing the bilayer number n. Therefore, it would not be fair if comparing the areal capacitance of thin (MXene/TAEA)_n multilayers with thick films that have thousands of layers such as pure MXene film (ref. 39) or MXene/polymer composites (ref. 21). We believe in this case the volumetric capacitance is more accurate as this value is mostly independent of mass loading. For example, the volumetric capacitance of (MXene/TAEA)₆ is as high as 583 F cm⁻³ at a scan rate of 2 mV s⁻¹.

I-5. It is mentioned, “cycling stability with 89% capacitance retention after 5,000 cycles at a current density of 0.1 mA cm⁻²” is an excellent value. For a supercapacitor this is a below average value!

Response: It has been reported by others that MXene Ti₃C₂T_x can be slowly oxidized during cyclic test by oxygen trapped in the aqueous electrolytes [ref. 39]. This suggested that cycling stability can be improved by removing the trapped oxygen from the electrolyte. In the revised manuscript, we therefore have repeated the capacitance retention test. When fabricating the devices and before running the experiment, we employed a vacuum-assisted degassing process to remove the trapped oxygen from the electrolyte as much as we can. These tests show a capacitance retention of about 90.3% after 10,000 cycles. We have added this new data as Supplementary Fig. 21 in the Supplementary Information.

“**Supplementary Fig. 21 | Cycle stability (MXene/TAEA)₂₀-based solid-state supercapacitors.** Insert is Nyquist plots of (MXene/TAEA)₂₀-based solid-state supercapacitors before and after 10 000 cycles. After long cycling our device shows a small increase in equal series resistances.”

We have also revised the text on [page 18] in the revised manuscript:

“...then they were treated at vacuum condition for overnight to vaporize the excess water and to remove the trapped oxygen from the electrolyte.”

and the text on [page 11] in the revised manuscript:

“Additionally, the solid-state supercapacitors based on (MXene/TAEA)₂₀ exhibited a cycling stability with 90.3% capacitance retention after 10,000 cycles (Supplementary Fig. 21).”

I-6. The discussion of the results and the reasons for enhanced performance is very weak. It is mentioned that the interlayer spacing is increased without giving any surface area values. Also, EIS of electrodes with and without TAEA is not compared. This is important to understand the mechanism of charge transfer in a pseudocapacitive environment.

In summary, the authors developed a multilayer MXene and TAEA and claim that it is an excellent material for supercapacitors when the actual electrochemical performances are below average. I do not recommend the publication of this paper in Nature Communications.

Response: We thank the reviewer for the comment. We would like to first explain and clarify that the most novel aspect of our work is the development of a novel LbL self-assembly process using small molecules to fabricate 2D multilayers with single/few layer precision in per bilayer over large areas and on different substrates, which is fundamentally different from LbL self-assembly with conventional polymers. Our success in the developing this method has been proven with the experiments and conclusive results. Moreover, the (MXene/TAEA)_n multilayers show the enhanced performances such as on the conductivity and capacitance compared with all other reported MXene-based multilayers fabricated by LbL self-assembly. In the revised manuscript, we have added much comprehensive comparison for the conductivity (see more response 1-3) and capacitive performance (see more response 1-2) of the (MXene/TAEA)_n multilayers with those of polymer-based multilayer such as (MXene/PEI)_n at the same bilayer number to confirm these points, and also compared the results with other reported polymer-based MXene multilayers. As explained in our responses to the previous comments, the addition of TAEA between the MXene layers enhances the ion diffusion between the layers. However, since the TAEA molecules are small, the increase in the interlayer spacing does not harm the conductivity of the electrodes. In other words, the expansion of interlayer spacing is enough to enhance the intercalation of protons but not too much to significantly decrease conductivity as it is observed for other LbL films. Regarding the EIS experiments, we have added new data to compare the EIS of (MXene/TAEA)₂₀ and (MXene/PEI)₂₀ electrodes in Supplementary Fig. 19. The (MXene/TAEA)₂₀ showed smaller equivalent series resistance than (MXene/PEI)₂₀.

Reviewer 2

The authors report on a new layered by layered stacking of 2D sheets of MXene using TAEA to control the spacing between MXene sheets and improve their bonding. Further, they showed the possibility of making all-solid-state supercapacitors with high volumetric capacitance and its mechanical stability. The work is novel at the level of Nature Comm. and

2D community and MXene community, in particular, can benefit from this work. However, a major revision is needed:

2-1. The bilayer stacking, interlayer spacing, and a MXene layer thickness need clear definitions. For example in “based on the thickness of single MXene flake (~ 2.5 nm), and the interlayer spacing (~1.4 nm) and the bilayer thickness (4.5 nm) of LbL MXene films” the interlayer spacing from XRD includes the thickness of a MXene flake (~1nm), so the gallery between the layers is only 0.4 nm. The bilayer thickness needs to be recalculated.

Response: We thank the reviewer for the comments. To avoid possible misunderstandings, we have further clarified the definition of bilayer by adding the text in the Methods section on [page 15] in the revised manuscript:

“The treated substrates were first dipped into TAEA solution for 5 min and were then rinsed 3 times by Milli-Q water for 3, 2, 1 min per time to remove the weakly absorbed molecules. After that, the cation-coated substrates were dipped into the $\text{Ti}_3\text{C}_2\text{T}_x$ MXene dispersion for 5 min and then rinsed with Milli-Q water as the same steps above. This cycle made one bilayer of $(\text{MXene/TAEA})_1\dots$ ”,

We have stated the definition of interlayer spacing for multilayer films from XRD data in Eq. (1) in the Method section [page 16 of the revised manuscript], the thickness of individual MXene flake from AFM height images [page 5 of the revised manuscript] and the thickness of $(\text{MXene/TAEA})_n$ multilayers from corresponding cross-sectional SEM images on [page 6 of the revised manuscript]. We also clarified the definition of average bilayer thickness in the text on [page 6] in the revised manuscript:

“An average increase of around 4.5 nm per bilayer was calculated from the thickness of multilayers divided by corresponding bilayer number n.”

2-2. In “The expansion of the interlayer spacing indicates that TAEA molecules have intercalated into the adjacent MXene flakes ...” It is not clear how the LbL would cause intercalation. It is more of stacking molecules over MXene by separate dip coating (LbL) than intercalation

Response: We understand the confusion in terminology. When we write “intercalated”, we refer to the LbL process in which positively charged TAEA molecules pillared onto the negatively charged surfaces of adjacent MXene flakes, through electrostatic interaction. To avoid confusion, we now used the word “pillared” instead of “intercalated” throughout the text.

2-3. Since TAEA and MXene are oppositely charged, have authors considered mixing the two solutions to create self-assembly formation of this hybrid structure. If it is not possible, a clear explanation is needed.

Response: The main point of this work is to show that we can control the growth of 2D multilayer films with single/few flake precision in each bilayer over large areas and various dimensional substrates. This was the main reason for the invention of the LbL self-assembly technique. When a positive and negative colloid are mixed, they just form self-assembled agglomerates and cannot create such structures. To further clarify this phenomenon, we added Supplementary Fig. 3 in the Supplementary Information to show the formation of MXene/TAEA agglomerates in solution:

“Supplementary Fig. 3 | Self-assembly formation of MXene/TAEA hybrids after mixing the MXene dispersion with TAEA solution. The photographs are for 1 g/L MXene dispersion, and the MXene/TAEA hybrids at the different time after mixing, from left to right.”

We further revised the text on [page 6] in the revised manuscript:

“The increased charge density is in favor of the self-assembly process (Supplementary Fig. 3), especially, LbL self-assembly as it provides stronger electrostatic interactions between TAEA and MXene.”

2-4. Authors say “ordered structures in LbL MXene films arise only when the counter phase is the small-molecule TAEA” What is the limit on this “small” molecule size (in nm)? More specific details.

Response: The discovery of proper molecules for LbL self-assembly of 2D materials with ordered structures was a big challenge. We have solved this problem by systematically studying ionic polymers and small molecules. We included some of these results in Supplementary Fig 4 in the Supplementary Information:

“**Supplementary Fig. 4 | LbL self-assembly of MXene flakes using different cationic counter phases.** This figure clearly shows that diamino small-molecule such as spermidine does not work for this LbL process. While the triamino molecules of TAEA and tris(3-aminopropyl)amine (TAPA) both work well. We therefore inferred that the triamino molecule is the minimum requirement that can lead to LbL self-assembly, and the TAEA is the smallest triamino molecule that can be chosen for LbL self-assembly process.”

To further clarify this point, we have added the following text to the revised manuscript on [page 6] in the revised manuscript:

“We also used other small amino molecules such as diamino molecule of spermidine, and slightly larger triamino molecule of tris(3-aminopropyl)amine (TAPA) as the counter phase to LbL self-assembly of MXene (Supplementary Fig 4). The results

showed that the triamino molecule is the minimum requirement for LbL self-assembly, and importantly the TAEA is the smallest triamino molecule which can be chosen for our LbL process.”

Additionally, we compared the MXene multilayers self-assembled by small molecule (TAEA) and polymer (PEI) in Supplementary Fig. 9 in the Supplementary Information, and revised the text on [page 7] in the revised manuscript:

“We note that a more ordered and smooth LbL structure was obtained only when the counter phase was the small-molecule TAEA, compared with those of polymer-based LbL structures such as (MXene/PEI)_n (Supplementary Fig. 9) and other reported (MXene/PDAC)_n²⁵ and (MXene-PVA/CNT-PSS)_n²⁶ even whose (0002) peak totally disappeared due to less uniform. We attributed the ordered structure to the small size of TAEA which forms a sub-nanometer gap in-between MXene flakes in the LbL films, leading to a quasi-intimate interfacial contact similar to pure MXene films.”

2-5. More details on the AFM and XRD of MXene TAEA is needed. Authors need to explain why there is no height change after TAEA in AFM and then there is a slight increase in the stacking of the flakes in XRD. A discussion about the exact size of the TAEA is needed.

Response: The AFM height profiles (Fig. 1c and f) are both for individual MXene flakes. To clarify this point, we have revised the text on [page 7] in the revised manuscript::

“Fig. 1f shows that the edge height of individual MXene flakes on the top layer of multilayers is ~ 2.5 nm, which is consistent with the thickness of individual flakes of pristine Ti₃C₂T_x MXene (Fig. 1c). This suggested that individual MXene flakes did not agglomerate during LbL process.”

and the text in Fig 1 caption on [page 20] in the revised manuscript:

“... and (f) corresponding height profiles at the edge of MXene flakes on the top layer of multilayers in (e) that were stacked on the next TAEA layer.”

The XRD patterns show an increase of average interlayer spacing for the entire (MXene/TAEA)_n multilayers compared with pure MXene film, caused by the pillaring effect of TAEA molecule during LbL process. To clarify the analysis for XRD data and the effect of the size of TAEA, we have modified the following text on [page 7-8] in the revised manuscript:

“Additionally, (0002) peaks of (MXene/TAEA)_n shifted from 6.89° for pure MXene films to 6.38°, which showed a uniformly pillaring effect of TAEA in LbL process.³⁸ This corresponds to an increase of 1.0 Å in the average interlayer spacing from 12.8 Å up to 13.8 Å. We note that this is a very small increase in interlayer spacing even if we consider the small size of the TAEA molecule, and results from complex interface interactions between MXene flakes and TAEA in a process which is yet not fully characterized and understood in this or other reported polymeric systems (e.g. MXene/PVA,²⁰ MXene/CTAB,³⁸ or MXene/PANI films²¹).”

2-6. In “small gap between the interface contact of MXene flakes resulting in high electronic conductivity,” It is not clear why the gap between MXene flakes helps the conductivity.

Response: What we claim here is that the small gap in the (MXene/TAEA)_n multilayers leads to a high conductivity compared with polymer-based MXene multilayers fabricated by LbL self-assembly. We have elaborated more on this point in details in response 1-3.

2-7. In “This conductivity is close to that of pure Ti₃C₂T_x MXene” the conductivity values are certainly lower than the values for MXene papers. More recent refs are needed. However, authors can mention their conductivity is similar to the MXene made in similar conditions (sonications, flake size, etc.).

Response: We have revised the text on [page 8] in the revised manuscript based on this comment:

“This conductivity is similar to that of films fabricated using small Ti₃C₂T_x flakes ($7.78 \times 10^5 \text{ S m}^{-1}$)³⁹ and Ti₃C₂T_x MXene clay ($1.5 \times 10^5 \text{ S m}^{-1}$)¹⁶ which were made by

similar methods to those used here, i.e., combined the etching by LiF+HCl solution and delamination by sonication.”

2-8. Authors mention: “This value is notably lower than that of polymer-based LbL MXene films: $830 \Omega \text{ sq}^{-1}$ for $(\text{MXene}/\text{PEI})_{20}$ ” relating it to the big size of molecules. To be more specific, XRD of MXene/PEI is needed to show the effect of the large PEI molecules.

Response: As per reviewer’s request, we have included XRD patterns of $(\text{MXene}/\text{PEI})_n$ multilayers in the Supplementary Information (supplementary Fig. 9), and elaborated more on this point in our response 1-3.

2-9. Why does the MXene/TAEA planar flat in fig 4a has a curvature? Was there any plasticity?

Response: This figure indeed shows $(\text{MXene}/\text{TAEA})_n$ multilayers on a plastic PET that is bent to show how the resistance of multilayers behaves under bending.

2-10. In the electrochemical section “The expanded interlayer spacing accelerates ion movement with a small charge transfer resistance”. However, the increase in the interlayer spacing is not significant ($\sim 0.4 \text{ \AA}$). More explanation is needed here and comparison with previously reported interlayer spacing of MXene.

Response: Based on the XRD results, the LbL films indeed showed an average interlayer spacing increase of 1 \AA (not 0.4 \AA). This increased interlayer spacing does not adversely affect the conductivity, but it is large enough to further facilitate the intercalation of very small electrolyte ions (protons here) during the electrochemical process. We have previously reported that even a small increase in the interlayer spacing through the incorporation of polyaniline layers between $\text{Ti}_3\text{C}_2\text{T}_x$ MXene layers can significantly improve the ion intercalation, reduce the ion diffusion resistance, and therefore improve the electrochemical performance of the MXene electrodes [ref. 21]. We believe similar mechanism governs LbL MXene/TAEA films, with the advantage that here we have control over the stacking and placement of the individual MXene layers, where such precision was not possible in our

previous work on solution phase deposition of polymers on MXenes. We also revised the text on [page 12] in the revised manuscript:

”The expanded interlayer spacing improves ion movement with a small charge transfer resistance (R_{ct}) of 2.3Ω obtained from electrochemical impedance spectroscopy (EIS) (Supplementary Fig. 17f), and provides more ion-accessible surface for redox reactions and interlayer volume to accommodate more electrolyte ions for charge storage.²¹”

Minor issues:

Full detail of the sequential LbL is lacking.

Response: This detail was included in Method in the revised manuscript.

Can authors make the zeta potential axis normal (increasing in the y-axis)

Response: Here, in this Figure, our intention was to show the change in absolute values of zeta potential as a function of pH.

Figure 2f. It is not clear why the Si substrate changes color from dark to bright gray by increasing the #n, from left to right.

Response: We have revised the digital photograph of the (MXene/TAEA)_n on silicon wafers, and included the digital photograph of the (MXene/TAEA)_n on glass slides in Figure 2f in the revised manuscript. We also added the text on [page 21] in the revised manuscript:

“(f) Digital photographs of the (MXene/TAEA)_n on silicon wafers, PET sheets and glass slides vs. n. The color change on Si wafers is owing to different optical characteristics of various thickness of (MXene/TAEA)_n.”

The term “bilayer” is not very clear. Does it mean a layer of MXene with TAEA or 2 layers of MXene with TAEA?

Response: Bilayer means two layers composed of one layer of MXene and one layer of TAEA, i.e., (MXene/TAEA)₁. We have elaborated on this point in Method section and our response to reviewer 2-1.

The term “overlapped face-to-face” is not clear. Does this mean stacking of the flakes?

Response: Yes, it means stacking. We have changed “overlapped” to “stacked” throughout the article.

Reviewer 3

The topic is very interesting. The paper is well written, organized, the figures number is fine. However, before publication in Nature Communication as free open access paper, a careful check of English language should be made in the interest of the authors, who diversely could lost the right interest from a broad readers. Therefore, I strongly suggest to revise the English language.

Response: We thank the reviewer for the comment. We have further revised the text of the revised manuscript, by a native English speaking person and ourselves.

*3-1. In Page 7, Line 158-161 and in the figure 2b, “This is the most remarkable result of this work since it suggests that each bilayer contains only 1-2 single MXene flakes, based on the thickness of single MXene flake (~ 2.5 nm), and the interlayer spacing (~1.4 nm) and the bilayer thickness (4.5 nm) of LbL MXene films.” In this sentence, the thickness of bilayer(4.5nm) was from the average with multiple layers of thickness which is not according with the signal flake
If the compositional uniformity and thickness flatness of the intermediate portion and the edge of this multilayer structure could be described in detail, it would be easier to explain the distance between the layers and explain the hydrophobicity characteristics.*

Response: We thank the reviewer for the comment. We have elaborated the definition of the thickness of individual MXene flakes, a bilayer, and multilayers, and the average interlayer spacing in responses 2-1 and 2-5.

Regarding the compositional uniformity and flatness of the multilayers, AFM height images (Fig. 1e, Supplementary Fig. 7 and 8) and top-view/cross-sectional SEM images

(Supplementary Fig. 5 and 6, and Fig. 2b) both showed the uniform and flat structures. The corresponding description in detail was shown on [page 7] in the revised manuscript.

“We further used AFM to analyze the in-plane microstructure of the LbL films (Fig. 1e, and Supplementary Fig. 7-8). The AFM images showed that all individual MXene flakes in multilayer films stacked face-to-face, in agreement with the top-view SEM images (Supplementary Fig. 5), with a very small arithmetical-mean-deviation roughness R_a of 2.45 nm in an area of $5 \mu\text{m} \times 5 \mu\text{m}$ (Supplementary Fig. 7).” “We propose that the uniformly face-to-face deposition of single MXene flakes results from the spin-assisted process which provides the shear force at the interface between MXene flakes and the substrate surface to prevent the deposition of large flakes and formation of thick agglomerates.³⁵”

3-2. What is the relationship between alternate completeness of two kind of layer materials and the capacitive properties of the pillared multilayers of $(\text{MXene/TAEA})_n$?

Response: If we understand this question correctly the reviewer refers to building LbL structures with alternating 2D materials rather than just one 2D material. This points to a very important implication of this work. We are currently exploring other 2D materials using similar LbL methods towards 2D heterostructures. This is a topic for further publication and we indeed think that this LbL method is of fundamental importance for achieving heterostructures on large scale.

3-3. In Figure 7 (sup), the difference in the contact angle among $n3.5$ and $n10$, $n30$ is not very obvious, does it mean that the two kind of layer materials are not completely alternated in the pillared multilayers of $(\text{MXene/TAEA})_n$?

Response: The thickness and mass loading of $(\text{MXene/TAEA})_n$ both showed a linear increase with the bilayer number n . This therefore indicated that the two materials are completely alternated during LbL self-assembly process that is consistent with our reported LbL work [ref 34]. To clarify this point, we added the text on [page 6] in the revised manuscript:

“The linear behavior is a feature of a successful LbL self-assembly,³⁴ and indicates that the two different materials completely alternate during the LbL process.”

To better explain their wettability, we have moved all contact angles of multilayers into Supplementary Fig. 10, and added the following text to the figure in the Supplementary Information:

“Supplementary Fig. 10 | Contact angles of MXene and (MXene/TAEA)_n with different bilayer numbers. n means the MXene as the last layer of the multilayers, while n.5 is the TAEA as the last layer. The photographs were taken after 20 s resting the water droplet on the surfaces.

It is noteworthy that the hydrophilicity of MXene multilayers increased with the multilayer number n, which was ascribed to the increase of surface roughness. While the multilayers finished with the TAEA as the last layer showed smaller contact angles than those of the multilayers finished with the MXene, suggesting TAEA layer with higher wettability. But the differences went down with the increase of bilayer number. The reason could be attributed to the very thin TAEA layer, and the wettability was dominated by the main building block of MXene flakes when increasing the thickness of multilayers.”

Reviewers' comments:

Reviewer #1 (Remarks to the Author):

It has already been reported many times that LBL technique can be used for supercapacitors. In addition, the LBL technique has been demonstrated for MXene too (Nanoscale, 2018, 10, 6005-6013). The areal capacitance in this paper (Nanoscale, 2018, 10, 6005-6013) is even better than that reported by authors. The major novelty of this paper is a high degree of precision with respect to the number of layers of MXene used. The other novelty is that the layers are easy to make. I still believe that this paper is not novel enough to publish in Nature Communications.

Reviewer #2 (Remarks to the Author):

The authors have answered my comments one by one and revised the manuscript accordingly and the paper can be accepted after a minor revision.

Although they have fixed the issue of bilayer thickness in the XRD, there is still a minor issue remaining: The proper method to measure the layer thickness is XRD, since it measures the average thickness and microscopy measure 1-2 spots. The SEM measurement is not accurate at the atomic level. The values from XRD and Braggs eq. should be used, so the thickness of a bilayer is ~ 1.38 nm. Authors need to revise the paper and only use XRD values instead of the 4.5 nm from AFM/SEM.

After fixing this comment, the paper can be accepted.

Point to Point Response to the Reviewers

Reviewer #1 (Remarks to the Author):

It has already been reported many times that LBL technique can be used for supercapacitors. In addition, the LBL technique has been demonstrated for MXene too (Nanoscale, 2018, 10, 6005-6013). The areal capacitance in this paper (Nanoscale, 2018, 10, 6005-6013) is even better than that reported by authors. The major novelty of this paper is a high degree of precision with respect to the number of layers of MXene used. The other novelty is that the layers are easy to make. I still believe that this paper is not novel enough to publish in Nature Communications.

Response:

We thank the reviewer for their comment, and respectfully ask them to consider our response in regard to their concern below.

We first want to point out that our paper never claims using layer-by-layer (LbL) technique to fabricate supercapacitor electrodes as its novelty. We agree with the reviewer that the novelty of our paper is partly due to the highly precise LbL self-assembly method where, for the first time, we have used a small molecule to LbL self-assemble 2D MXene flakes and we obtained highly ordered MXene multilayered structures with precise sub-nanometer d-spacing between each bilayer. We also have fabricated the MXene multilayers on several different non-conductive substrates to show the versatility of our approach and the flexibility of the fabrication method. However, as explained below this is not the only novel aspect of our work.

Other novel aspects of our work:

We also would like to point out to some of the other novel aspects of our work that demonstrate a high degree of precision in the LbL self-assembly process beyond what has been reported to date in the literature. **Our manuscript significantly contributes to the advancement of the field of self-assembly of MXenes and other 2D materials by showing that there is a minimum size limit for the counter phase for the LbL self-assembly of 2D materials. We have used the smallest triamino molecule of tris(2-aminoethyl) amine (TAEA) that can yield a successful LbL assembly as shown in Supplementary Fig. 4. The fabricated LbL MXene multilayers possess highly ordered structures, and show the integrated multifunctionality with the highest electronic conductivity, the highest volumetric capacitances, and the best resistance to extreme mechanical deformations**

(i.e., bending, knotting and compressing), compared to all of the previously reported MXene LbL multilayers using polymers as the counter phases, i.e., MXene/poly(ethyleneimine) (PEI),¹ MXene/PEI-modified CNT (*Nanoscale*, 2018, 10, 6005-6013),² MXene/poly(diallyldimethylammonium chloride) (PDAC),³ and poly(vinyl alcohol) (PVA)-modified MXene/poly(sodium 4-styrene sulfonate) (PSS)-modified CNT.⁴ This is because of the small size and high charge density of TAEA molecule we used in the LbL process.

During the LbL self-assembly process, TAEA creates a thin, highly uniform, sub-nanometer layer on top of the 2D flakes and its multiple charged groups reverse the surface charge of 2D flakes without disrupting their atomically flat surfaces. These features enable fabrication of MXene multilayers with a linearly growing thickness with single/few nanosheets in each bilayer, indicating a precise control over uniform stacking of MXene flakes in each bilayer. This behavior is further evident from the XRD data (Figure 2e) where we have shown that the fabricated LbL multilayers are very ordered with sharp and distinct XRD peaks corresponding to basal planes of MXenes, similar to that observed for MXene films produced with vacuum filtration method. Freestanding films of MXene (and other 2D materials) fabricated through vacuum filtration are known to have a high degree of orientation as the result of the vacuum-assisted orientation of flakes. However, such highly ordered structures have not been shown for any reported MXene LbL multilayers, (or indeed any other 2D system fabricated through LBL method), where using other polymer molecules as counter phase results in random orientation of flakes and lack of control over their stacking number in each bilayer. Because of the smaller gap between their MXene flakes and their highly ordered structures, the conductivity of the MXene LbL multilayers reported in our manuscript is significantly higher than those reported in the previous MXene LbL papers and is comparable to the conductivity of MXene films fabricated by vacuum-assisted filtration (see Figure 2d). The high charge density of TAEA also enhances the electrostatic interactions between MXene flakes leading to better electromechanical properties under extreme bending, knotting, and compressing conditions (Figure 4). For example, the MXene/TAEA multilayers we fabricated in this research, showed a resistance increase of only 1.2 times after 100 bending cycles and 1.5 times after 1000 cycles, which is significantly lower than the previously reported MXene LbL multilayers (*Nanoscale*, 2018, 10, 6005-6013) which showed a 2.2-3.9 times higher resistance after 100 bending cycles.

Supercapacitor performance

We would like to address the reviewer's criticism about the areal capacitance of the LbL MXene multilayers reported in our manuscript. Our response will clearly show that the LbL MXene multilayers reported in our manuscript are not commensurable with the ones reported in the *Nanoscale* paper (*Nanoscale*, 2018, 10, 6005-6013).²

We have used two metrics of areal capacitance and volumetric capacitance to report the specific capacitance of the electrodes to be consistent with other publications in the area of thin-film supercapacitor electrodes. Here, we explain the difference between each metric and compare our result to relevant literature reports. At the end of this section, we explain why the performance reported in the paper mentioned by the reviewer is not relevant and cannot be compared with the performance reported in our manuscript.

Areal capacitance

The areal capacitance, which is the reviewer's concern, is calculated by normalizing the capacitance of the electrode by its surface area and therefore, it is dependent not only on the performance of the electrode material but also on the amount of electrode material loaded in the unit area of the electrode (the so-called mass loading of the electrode). In other words, for thin film electrodes, the areal capacitance is significantly affected by the electrode thickness. In fact, Figure 5f in our paper very nicely demonstrates this effect as it shows that by increasing the thickness of the electrode (number of bilayers) the areal capacitance of the electrode consistently increases. By increasing the bilayer number from 6 to 20, we see about a 5-fold increase in the areal capacitance. Therefore, an accurate and fair comparison of the areal capacitance of the reported electrodes should only be made with consideration of the electrode mass loading and thickness. In our manuscript, we report an areal capacitance of about 4.8 mF cm² for an electrode with 20 bilayers (thickness of about ~89 nm). In our review of the literature, we found two other reports that MXene film electrodes with similar thicknesses were used and areal capacitance was reported. Both papers have been recently published in high impact journals. In the first paper (*Adv. Mater.* **2017**, 29 (36) 1702678), an areal capacitance of 1.6 mF cm⁻² was reported for an electrode with a thickness of ~88 nm. Please take a look at Figure 3c of this paper, where the maximum calculated capacitance of electrodes with a thickness of about 150 nm is only 3.2 mF cm⁻².⁵ In the second paper (*Adv. Funct. Mater.* **2018**, 28 (9), 1705506.), when the thickness of the MXene electrodes was about 90 nm, a maximum areal capacitance of 2.5 mF cm⁻² was achieved.⁶ Therefore, a fair

comparison of the reported areal capacitances, clearly shows that the electrodes reported in our manuscript are among the best performing thin film MXene electrodes. This is also confirmed by the high volumetric capacitance of the electrodes reported in our manuscript.

Volumetric capacitance

Volumetric capacitance is calculated by dividing the total capacitance of the electrode by its volume and is considered to be a better metric to compare the performance of various electrodes.⁷ In our work, we have reported a volumetric capacitance of 583 F cm⁻³ for the LbL MXene multilayers in practical solid-state devices (Figure 5g). This is attributed to the ordered assembly of MXene flakes during our LbL process creating a highly packed (dense) structure of the assembled MXene flakes that results in a high volumetric performance.

Comparison with the paper referred to by the reviewer (Nanoscale 2018, 10, 6005-6013):

We would like to point out several major differences between our work and the work reported in the paper cited by the reviewer (Zhou et al., Nanoscale 2018, 10, 6005-6013). **Firstly, the capacitances of the electrodes reported by Zhou et al. are not only from MXene, but also from the combination of the whole composite electrode structure including the carbonized polycaprolactone fibers, carbon nanotube, and even nitrogen-doped carbon from the carbonization of PEI.** In this paper, the authors used PEI-modified CNT for LbL assembly of MXene flakes, and they actually can only conduct the LbL process under two bilayers. Due to the polymer PEI incorporated in LbL process, their obtained MXene multilayers have to undergo a post-carbonization step (500 °C for 2 hours) to carbonize the PEI and the substrates of polycaprolactone fibers, in order to improve the conductivity for the charge storage. **Secondly, the mass loading of active electrode materials reported by Zhou et al. is almost 10 times higher than the mass loading of the electrodes reported in our paper, because the porous substrate with high surface area can load more active material than planar substrate at certain areal.** Therefore, the higher areal capacitance reported by Zhou et al is mostly because they have more active electrode material in a unit area. While Zhou et al. have chosen to not report their electrode mass loading, the number can be easily estimated by dividing the reported areal capacitance (shown with the unit of mF cm⁻² in Fig. 5b in Zhou et al.) by the reported gravimetric capacitance (shown with the unit of F g⁻¹ in Fig. S7b). By using this method, one can calculate the mass loading of their best

performing electrode to be about 0.24 mg cm^{-2} . However, the mass loading of our best performing electrode is only about 0.029 mg cm^{-2} , which means that our electrodes have about 10 times less active material in the unit area. In Zhou et al., the MXene hybrid electrodes were fabricated on a porous fiber-based substrate that can also take more active material in a unit area (as mentioned above) than the planar substrate we used.

Reviewer #2 (Remarks to the Author):

The authors have answered my comments one by one and revised the manuscript accordingly and the paper can be accepted after a minor revision.

Although they have fixed the issue of bilayer thickness in the XRD, there is still a minor issue remaining: The proper method to measure the layer thickness is XRD, since it measures the average thickness and microscopy measure 1-2 spots. The SEM measurement is not accurate at the atomic level. The values from XRD and Bragg's eq. should be used, so the thickness of a bilayer is $\sim 1.38 \text{ nm}$. Authors need to revise the paper and only use XRD values instead of the 4.5 nm from AFM/SEM.

After fixing this comment, the paper can be accepted.

Response: We thank the reviewer for the constructive comments, and agree that XRD is the most accurate method to measure the average interlayer spacing between MXenes, which the reviewer refers to as “thickness of a bilayer”. Please note, however, that the definition of bilayer in the manuscript may be different from the definition used by the reviewer. The bilayer n is defined as the number of steps repeated in the LbL process, and not bilayer as in two layers of MXene or one layer of MXene and one layer of TAEA. The definition of bilayer was included in the methods section in our previous revision of manuscript. To further clarify this point we have also added the following text on [page 5]:

“The LbL films are denoted $(\text{MXene/TAEA})_n$ where n corresponds to the number of steps that are repeated in the LbL deposition process.”

We also wish to point out that the 2D flakes in the LbL films form random structures in plane and the relation between n and average film thickness, and average interlayer MXene spacing is indeed very complex, and subject to further studies.

Therefore, to avoid any further confusion between the concepts of bilayer, interlayer spacing, and film thickness, we have, based on the reviewer comments, removed the text [on page 6]:

“An average increase of around 4.5 nm per bilayer was calculated from the thickness of multilayers divided by corresponding bilayer number n.”

We have further added the following text [on page 7]:

“which means an average interlayer distance of 1.38 nm between the MXene flakes in the multilayer films.”

to use the correct value of the average spacing between adjacent MXene flakes as calculated from XRD data, and not the values estimated from SEM/AFM, based on the reviewers suggestion.

References

- 1 Tian Y. *et al.* Ni foam supported quasi-core-shell structure of ultrathin Ti₃C₂ nanosheets through electrostatic layer-by-layer self-assembly as high rate-performance electrodes of supercapacitors. *J. Power Sources* **369**, 78-86 (2017).
- 2 Zhou Z. *et al.* Layer-by-layer assembly of MXene and carbon nanotubes on electrospun polymer films for flexible energy storage. *Nanoscale* **10**, 6005-6013 (2018).
- 3 An H. *et al.* Surface-agnostic highly stretchable and bendable conductive MXene multilayers. *Sci. Adv.* **4**, eaaq0118 (2018).
- 4 Weng, G.-M. *et al.* Layer-by-layer assembly of cross-functional semi-transparent mxene-carbon nanotubes composite films for next-generation electromagnetic interference shielding. *Adv. Funct. Mater.* **0**, 1803360 (2018).
- 5 Zhang, C. J. *et al.* Transparent, Flexible, and conductive 2d titanium carbide (MXene) films with high volumetric capacitance. *Adv. Mater.* 1702678 (2017).
- 6 Zhang, C. *et al.* Stamping of flexible, coplanar micro-supercapacitors using mxene inks. *Adv. Funct. Mater.*, 1705506 (2018).
- 7 Gogotsi, Y. & Simon, P. True performance metrics in electrochemical energy storage. *Science* **334**, 917-918 (2011).